# Extraction and Recovery of Spatio-Temporal Structure in Latent Dynamics Alignment with Diffusion Models

**Yule Wang**
Georgia Institute of Technology
Atlanta, GA, 30332 USA
`yulewang@gatech.edu`

**Zijing Wu**
Georgia Institute of Technology
Atlanta, GA, 30332 USA
`zwu381@gatech.edu`

**Chengrui Li**
Georgia Institute of Technology
Atlanta, GA, 30332 USA
`cnlichengrui@gatech.edu`

**Anqi Wu**
Georgia Institute of Technology
Atlanta, GA, 30332 USA
`anqiwu@gatech.edu`

## Abstract

In the field of behavior-related brain computation, it is necessary to align raw neural signals against the drastic domain shift among them. A foundational framework within neuroscience research posits that trial-based neural population activities rely on low-dimensional latent dynamics, thus focusing on the latter greatly facilitates the alignment procedure. Despite this field's progress, existing methods ignore the intrinsic spatio-temporal structure during the alignment phase. Hence, their solutions usually lead to poor quality in latent dynamics structures and overall performance. To tackle this problem, we propose an alignment method ERDiff, which leverages the expressivity of the diffusion model to preserve the spatio-temporal structure of latent dynamics. Specifically, the latent dynamics structures of the source domain are first extracted by a diffusion model. Then, under the guidance of this diffusion model, such structures are well-recovered through a maximum likelihood alignment procedure in the target domain. We first demonstrate the effectiveness of our proposed method on a synthetic dataset. Then, when applied to neural recordings from the non-human primate motor cortex, under both cross-day and inter-subject settings, our method consistently manifests its capability of preserving the spatio-temporal structure of latent dynamics and outperforms existing approaches in alignment goodness-of-fit and neural decoding performance. Codes are available at: `https://github.com/alexwangNTL/ERDiff`.

## 1 Introduction

A key challenge severely impeding the scalability of behavior-related neural computational applications is their robustness to the distribution shift of neural recordings over time and subjects [1]. Given a behavior model trained on previous neural recordings (e.g., velocity predictor for human with paralysis [2]), it usually suffers performance degradation when applied to new neural recordings due to the neural distribution shift [3, 4]. Thus, for long-term usability and stable performance of the trained neural decoding model, high-quality alignment between the neural recordings used for training (i.e., source domain) and new recordings for testing (i.e., target domain) is of vital importance.

Distribution alignment is an important task at the heart of unsupervised transfer learning [5, 6]. The goal is to align the target domain to the source domain so that the trained model in the source domain can be applied to the target domain after eliminating the distribution shift. However, due to issues such as instabilities and low signal-to-noise ratio [7], raw neural spiking activities are noisy and ambiguous [8, 9], causing difficulties in aligning the distributions of these high-dimensional signals directly.

37th Conference on Neural Information Processing Systems (NeurIPS 2023).

One promising research direction [10] points out that the trial-based neural activities can always be understood in terms of low-dimensional latent dynamics [11, 12, 13]. Such latent dynamics manifest coordinated patterns of evolution constrained to certain "neural manifolds" [14, 15]. Hence, early studies focusing on the alignment of latent dynamics reach comparably satisfactory results [16, 17]. Generally, most previous methods [16, 18, 19] are based on a pre-defined metric for optimization during latent dynamics alignment, i.e., minimizing the difference evaluated by the metric, between source and target domains within the low-dimensional latent space. However, those metrics are usually non-parametric and handcrafted, which are not guaranteed to suit specific neural recordings or problems well. Methods based on adversarial-learning [20, 21] thus have been introduced since they can implicitly find an adapted metric [22]. However, they suffer from mode collapse and instability issues in practice [23].

Moreover, during the alignment process, we note that the above-mentioned works lack the necessary awareness of the latent dynamics structure, especially when aligning non-linear and lengthy trials. Through an empirical study on the motor cortex of non-human primate (NHP) [8] (shown in Figure 1), we can observe that: a state-of-the-art alignment method JSDM [24] (minimizing the symmetric Jensen–Shannon divergence between distributions) fails to recover the latent dynamics structures of the source domain since JSDM neglects those structures during alignment. From another perspective, in the alignment phase, existing methods fail to effectively model and leverage the information-rich correlations between each time bin and each latent dimension within latent dynamics.

In this paper, we focus on preserving the *temporal evolution* of each individual latent dimension and the *spatial covariation* between latent dimensions of the source domain during alignment. The main idea is that we first extract the spatio-temporal structure of latent dynamics from the source domain; and then, we align the target domain by recovering the source domain's underlying structure. However, such a workflow is non-trivial since the underlying spatio-temporal structure is both implicit and complex.

To tackle this problem, we propose a novel alignment method that is capable of *E*xtracting and *R*ecovering the latent dynamics structure with *Diff*usion model (ERDiff). Firstly, given the source-domain neural observations, we use a diffusion model (DM) [25, 26] to extract the spatio-temporal structure of latent dynamics. Then, in the alignment phase, we propose a maximum likelihood alignment procedure through the guidance of DM, by which the spatio-temporal structure of source-domain latent dynamics can be recovered well in the target domain. The proposed extract-and-recover method nicely encodes and preserves the spatio-temporal structure of latent dynamics, which are significant inductive biases for neural latent dynamics alignment. Furthermore, from the perspective

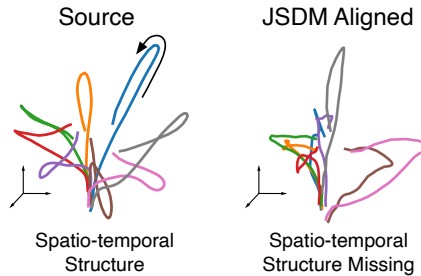

Source        JSDM Aligned

Spatio-temporal Structure      Spatio-temporal Structure Missing

Figure 1: **Empirical study**. Latent dynamics (3D visualization) of the source domain and the aligned target domain by JSDM on a primary motor cortex dataset.

of core machine learning, ERDiff introduces an approach of extracting structure knowledge from one distribution and imposing it as the prior to constrain the alignment of another distribution. Note that although we have been emphasizing extraction and recovery of the source-domain structure, ERDiff is not performing a copy-and-paste of the source domain distribution to the target domain. As ERDiff preserves the dynamics structure of the source domain, it also maintains the original characteristics of the target domain. We present experimental results to support this argument. Finally, we conduct extensive experiments to verify the effectiveness of ERDiff on a synthetic dataset and two real-world neural datasets [8, 27]. Visualization of latent dynamics also demonstrates that ERDiff is capable of preserving the spatio-temporal structure consistently during the alignment phase.

## 2 Preliminary

**Distribution alignment.** We denote the source-domain observations of single-trial neural population activities as $\mathbf{X}^{(s)} = \left[\mathbf{x}_1^{(s)}, \ldots, \mathbf{x}_l^{(s)}\right]^\top \in \mathbb{R}^{l \times n}$, where $l$ is the trial length (*i.e.*, number of time bins), and $n$ is the number of observed neurons. We denote its low-dimensional latent dynamics as $\mathbf{Z}^{(s)} = \left[\mathbf{z}_1^{(s)}, \ldots, \mathbf{z}_l^{(s)}\right]^\top \in \mathbb{R}^{l \times d}$, where $d$ is the latent dimension size. Generally, we build a variational autoencoder (VAE) to estimate the latent $\mathbf{Z}^{(s)}$ given the observations $\mathbf{X}^{(s)}$. The VAE

consists of a probabilistic encoder $q(\mathbf{Z}^{(s)} \mid \mathbf{X}^{(s)}; \boldsymbol{\phi}_s)$ and a probabilistic decoder $p(\mathbf{X}^{(s)} \mid \mathbf{Z}^{(s)}, \boldsymbol{\psi}_s)$. $\boldsymbol{\phi}_s$ and $\boldsymbol{\psi}_s$ are the parameters of the encoder and decoder. The encoder also serves as an approximated posterior distribution to the intractable true posterior $p(\mathbf{Z}^{(s)} \mid \mathbf{X}^{(s)})$. Then in the target domain, given the neural population activities $\mathbf{X}^{(t)} = \left[ \mathbf{x}_1^{(t)}, \ldots, \mathbf{x}_l^{(t)} \right]^\top \in \mathbb{R}^{l \times n}$, we perform distribution alignment by linear probing the probabilistic encoder $q(\mathbf{Z} \mid \mathbf{X}; \boldsymbol{\phi})$. This alignment phase is conducted by minimizing certain probability divergence $\mathbb{D}(\cdot \mid \cdot)$ between the two posterior distributions:

$$\min_{\phi_t} \mathbb{D}(q(\mathbf{Z}^{(s)} \mid \mathbf{X}^{(s)}; \boldsymbol{\phi}_s) \| q(\mathbf{Z}^{(t)} \mid \mathbf{X}^{(t)}; \boldsymbol{\phi}_t)). \tag{1}$$

**Diffusion (probabilistic) model (DM).** Given $l \times d$-dimensional i.i.d. samples $\mathbf{Z}$ from an unknown data distribution, a DM [28] aims to approximate such distribution by fitting the parameters of a neural network $p_{\boldsymbol{\theta}}(\mathbf{Z})$. DM is composed of a *forward process* followed by a *reverse process*. In the *forward process*, isotropic Gaussian noise is added to diffuse the original data, which can be defined in a linear stochastic differential equation (SDE) form:

$$d\mathbf{Z} = \boldsymbol{f}(\mathbf{Z}, t)dt + g(t)d\mathbf{w}, \tag{2}$$

where $\boldsymbol{f}(\cdot) : \mathbb{R}^{l \times d} \times \mathbb{R} \mapsto \mathbb{R}^{l \times d}$ is the drift coefficient, $g(\cdot) : \mathbb{R} \mapsto \mathbb{R}$ is the diffusion coefficient, and $\mathbf{w}$ is the standard Wiener process. The solution of the SDE is a diffusion process $\{\mathbf{Z}_t\}_{t \in [0,T]}$, in which $[0, T]$ is a fixed time zone. In this paper, we implement them with VP-SDE [25]. $\{\mathbf{Z}_t\}_{t \in [0,T]}$ approaches the standard normal prior distribution $\pi(\mathbf{Z})$ when $t = T$. Under mild conditions on drift and diffusion coefficients [25], the denoising *reverse process* can be solved in the following closed-form SDE:

$$d\mathbf{Z} = \left[ \boldsymbol{f}(\mathbf{Z}, t) - g(t)^2 \nabla_{\mathbf{Z}} \log p_t(\mathbf{Z}) \right] dt + g(t)d\overline{\mathbf{w}}, \tag{3}$$

where $\nabla_{\mathbf{Z}} \log p_t(\mathbf{Z})$ is the score function, and $\overline{\mathbf{w}}$ is a reverse-time Wiener process. We train a parameterized network $\boldsymbol{s}(\mathbf{Z}, t; \boldsymbol{\theta})$ to fit the score function $\nabla_{\mathbf{Z}} \log p_t(\mathbf{Z})$. However, $\nabla_{\mathbf{Z}} \log p_t(\mathbf{Z})$ is not directly accessible and we resort to the denoising score matching (DSM) [29] for optimization:

$$\mathcal{L}_{\text{DSM}}(\boldsymbol{\theta}) = \mathbb{E}_{t \sim \mathcal{U}[0,T]} \mathbb{E}_{\mathbf{Z}_0 \sim p, p_{0t}(\mathbf{Z}_t \mid \mathbf{Z}_0)} \left[ \lambda(t)^2 \| \nabla_{\mathbf{Z}_t} \log p_{0t}(\mathbf{Z}_t \mid \mathbf{Z}_0) - \boldsymbol{s}(\mathbf{Z}_t, t; \boldsymbol{\theta}) \|_2^2 \right], \tag{4}$$

where $\mathcal{U}$ represents the uniform distribution and $\lambda(t)$ is the weighting function. Under VP-SDE, the transition probability $p_{0t}(\mathbf{Z}_t \mid \mathbf{Z}_0)$ also follows a Gaussian distribution $\mathcal{N}(\mu_t \mathbf{Z}_0, \Sigma_t)$, in which $\mu_t, \Sigma_t \in \mathbb{R}^{l \times d}$. On the other hand, according to [28], we can define a noise estimator with the score function as $\boldsymbol{\epsilon}(\mathbf{Z}_t, t; \boldsymbol{\theta}) = -\boldsymbol{K}_t^{-T} \boldsymbol{s}(\mathbf{Z}_t, t; \boldsymbol{\theta})$, in which $\boldsymbol{K}_t \boldsymbol{K}_t^T = \Sigma_t$. Invoking these expressions, we can thus reformulate the form of DSM loss based on the Fisher Divergence between noise terms:

$$\mathcal{L}_{\text{DSM}}(\boldsymbol{\theta}) = \mathbb{E}_{t \sim \mathcal{U}[0,T]} \mathbb{E}_{\mathbf{Z}_0 \sim p, \boldsymbol{\epsilon} \sim \mathcal{N}(0, \boldsymbol{I}_{l \times d})} \left[ w(t)^2 \| \boldsymbol{\epsilon} - \boldsymbol{\epsilon}(\mathbf{Z}_t, t; \boldsymbol{\theta}) \|_2^2 \right], \tag{5}$$

in which $w(t) = \boldsymbol{K}_t \lambda(t)$ and $\mathbf{Z}_t = \mu_t \mathbf{Z}_0 + \boldsymbol{K}_t \boldsymbol{\epsilon}$.

## 3 Methodology

In this section, we introduce our proposed latent dynamics alignment method ERDiff in detail.

### 3.1 Maximum likelihood alignment

Given the source-domain neural activities $\mathbf{X}^{(s)}$, we infer their latent dynamics $\mathbf{Z}^{(s)}$ by building a VAE. We use variational inference to find the probabilistic encoder $q(\mathbf{Z}^{(s)} \mid \mathbf{X}^{(s)}; \boldsymbol{\phi}_s)$ and probabilistic decoder $p(\mathbf{X}^{(s)} \mid \mathbf{Z}^{(s)}; \boldsymbol{\psi}_s)$ through maximization of the evidence lower bound (ELBO) [30]:

$$\boldsymbol{\phi}_s, \boldsymbol{\psi}_s = \underset{\boldsymbol{\phi}, \boldsymbol{\psi}}{\operatorname{argmax}} \left[ \mathbb{E}_{q(\mathbf{Z}^{(s)} \mid \mathbf{X}^{(s)}; \boldsymbol{\phi})} \left[ \log p(\mathbf{X}^{(s)} \mid \mathbf{Z}^{(s)}; \boldsymbol{\psi}) \right] - \mathbb{D}_{\text{KL}} \left( q(\mathbf{Z}^{(s)} \mid \mathbf{X}^{(s)}; \boldsymbol{\phi}) \| \bar{q}(\mathbf{Z}^{(s)}) \right) \right], \tag{6}$$

in which $\bar{q}(\mathbf{Z}^{(s)})$ is the normal prior. Note that we introduce ERDiff with this basic VAE architecture. But ERDiff can be combined with many variants of latent variable models (LVM) [31, 32]. The essence of ERDiff is to tune the parameter set $\boldsymbol{\phi}$ of the probabilistic encoder, regardless of the model architecture of the encoder and decoder.

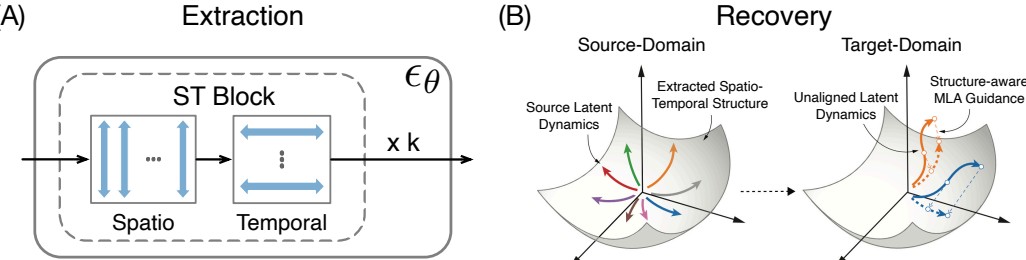

Figure 2: **A schematic overview of spatio-temporal structure extraction and recovery in ERDiff**. **(A)** The architecture of DM for spatio-temporal structure extraction. **(B)** A descriptive diagram of structure recovery schematic. The left presents the extracted spatio-temporal structure of the source-domain latent dynamics; the right illustrates the structure-aware maximum likelihood alignment guidance in ERDiff.

Practically, alignment methods that directly match the discrete samples from the source and target domains in a pair-wise fashion may lead to sub-optimal solutions [33, 34], especially when the collected samples from the target domain are limited. Thus given the target-domain neural activity $\mathbf{X}^{(t)}$, we propose to perform neural distribution alignment via maximum likelihood estimation (MLE):

$$\operatorname*{argmax}_{\phi} \mathbb{E}_{\mathbf{X}\sim p(\mathbf{X}^{(t)})}\left[\log p_s\left(h(\mathbf{X};\phi)\right)\right] = \operatorname*{argmax}_{\phi} \mathbb{E}_{\mathbf{Z}\sim q(\mathbf{Z}|\mathbf{X}^{(t)};\phi)}\left[\log p_s(\mathbf{Z})\right], \tag{7}$$

in which $p_s(\mathbf{Z})$ represents the ground-truth probabilistic density of latent dynamics in the source domain and $h(\cdot)$ refers to the non-linear transformation from $\mathbf{X}$ to $\mathbf{Z}$ underlying the probabilistic encoder $q(\mathbf{Z} \mid \mathbf{X};\phi)$. The objective in Eq. 7 implies that, instead of minimizing a distance metric between source observations and target observations, we aim at maximizing the likelihood where the density comes from the source domain and the data comes from the target domain. The left-hand-side (LHS) is the MLE for observation $\mathbf{X}$ and the right-hand-side (RHS) is the MLE for latent $\mathbf{Z}$. We will focus on the RHS in the following sections. We note that the RHS objective implies that we will optimize the encoder parameter $\phi$ during alignment so that the latent encoder will map $\mathbf{X}^{(t)}$ to a proper latent $\mathbf{Z}^{(t)}$ who fits the source density $p_s(\mathbf{Z})$ well.

### 3.2 Spatio-temporal structure extraction and source domain learning

In order to calculate the objective function in Eq. 7, we need to know two density functions: $q(\mathbf{Z} \mid \mathbf{X};\phi)$ is defined in the original VAE model with the learnable parameter $\phi$; $p_s(\mathbf{Z})$ is the density of latent $\mathbf{Z}$ for the source domain. The latter is inaccessible by building a VAE alone. Therefore, the first step is to learn $p_s(\mathbf{Z})$ given only $\mathbf{X}^{(s)}$. We propose to learn $p_s(\mathbf{Z})$ through training a DM.

To fully capture $p_s(\mathbf{Z})$, the DM should consider the overall spatio-temporal structure of latent dynamics. To extract such a structure, the DM can not treat each latent state or time bin within latent dynamics as mutually independent and feed them into the model sequentially. We thus take the entire trial of latent dynamics $\mathbf{Z}_0^{(s)} \sim q(\cdot \mid \mathbf{X}^{(s)};\phi_s)$ as input to the DM for training. Specifically, the DM fits $p_s(\mathbf{Z})$ through the training of a denoiser $\boldsymbol{\epsilon}(\mathbf{Z},t;\boldsymbol{\theta}_s) : \left(\mathbb{R}^{l\times d} \times \mathbb{R}\right) \to \mathbb{R}^{l\times d}$.

Next, we describe the architecture of $\boldsymbol{\epsilon}(\mathbf{Z},t;\boldsymbol{\theta}_s)$, which is refined for extracting the global spatio-temporal structure of latent dynamics. Traditional architecture based on 2D-Convolution Layers [35] focuses on capturing the local features within latent dynamics, which can hardly extract its global spatio-temporal dependency or structure. Thus, we adopt an architecture mainly derived from Diffusion Transformer (DiT) [36, 37]. Specifically, we propose to use *Spatio-Temporal Transformer Block* (STBlock), shown in Figure 2(A). Each STBlock is composed of a Spatio Transformer layer followed by a Temporal Transformer layer, which are 1-layer encoders based on multi-head self-attention. The Spatio Transformer layer takes latent states of each time bin as inputs to extract spatial structure, whereas the Temporal Transformer layer takes the entire latent trajectory of each latent space dimension as inputs to extract temporal structure. (see Appendix A for details of the architecture of DM).

For the training objective of $\epsilon(\cdot; \boldsymbol{\theta}_s)$, we sample noisy targets $\mathbf{Z}_t^{(s)}$ and minimize the following DSM loss function:

$$\boldsymbol{\theta}_s = \underset{\boldsymbol{\theta}}{\operatorname{argmin}} \, \mathbb{E}_{t \sim \mathcal{U}[0,T]} \mathbb{E}_{\mathbf{Z}_0^{(s)} \sim q(\cdot | \mathbf{X}^{(s)}; \boldsymbol{\phi}_s), \epsilon \sim \mathcal{N}(\mathbf{0}, \mathbf{I}_{l \times d})} \left[ w(t)^2 \left\| \epsilon - \epsilon(\mathbf{Z}_t^{(s)}, t; \boldsymbol{\theta}) \right\|_2^2 \right]. \quad (8)$$

We note that $\mathbf{Z}_0^{(s)}$ here are actually latent dynamics inferred via VAE in Eq. 6. To enrich the input samples and adequately estimate the source density $p_s(\mathbf{Z})$ as motivated earlier, we propose to learn the VAE objective (Eq. 6) and the diffusion objective (Eq. 8) simultaneously. In each training iteration, conditioning on the current value of $\boldsymbol{\phi}_s$ and $\boldsymbol{\psi}_s$, we obtain a set of $\mathbf{Z}_0 = h(\mathbf{X}^{(s)}; \boldsymbol{\phi}_s)$ and use it as $\mathbf{Z}_0^{(s)}$ to optimize Eq. 8. We can also optimize VAE first, obtain an optimal $\boldsymbol{\phi}_s$, and use it to optimize Eq. 8. Experimental results show that the former approach achieves higher density estimation performance compared to the latter (see Appendix A for details).

### 3.3 Spatio-temporal structure recovery and distribution alignment

Given the trained denoiser $\epsilon(\mathbf{Z}, t; \boldsymbol{\theta}_s)$, we go through the reverse process from $t = T$ to $t = 0$ in Eq.(3) and obtain the marginal distribution $p_0(\mathbf{Z}; \boldsymbol{\theta}_s)$. We use $p_0(\mathbf{Z}; \boldsymbol{\theta}_s)$ to approximate $p_s(\mathbf{Z})$ in Eq. (7). The maximum likelihood estimation can thus be written as

$$\underset{\boldsymbol{\phi}}{\operatorname{argmax}} \, \mathbb{E}_{\mathbf{Z} \sim q(\mathbf{Z} | \mathbf{X}^{(t)}; \boldsymbol{\phi})} \left[ \log p_0(\mathbf{Z}; \boldsymbol{\theta}_s) \right]. \quad (9)$$

We perform alignment by tuning the parameter set $\boldsymbol{\phi}$ of the probabilistic encoder while keeping the DM $p_0(\mathbf{Z}; \boldsymbol{\theta}_s)$ fixed. Note that we have already optimized the VAE objective to obtain an optimal $\boldsymbol{\phi}_s$ using source data. During alignment, we first set $\boldsymbol{\phi}$ as $\boldsymbol{\phi}_s$ and then linear probe $\boldsymbol{\phi}$ (e.g., neural observation read-in layer). Consequently, we not only initialize the model with a good encoder but also make optimization during alignment much faster and more efficient.

In the reverse process, the computation of $\log p_0(\mathbf{Z}; \boldsymbol{\theta}_s)$ is tractable through the probability flow ODE [25] whose marginal distribution at each time step $t$ matches that of our VP-SDE. However, the direct computation of $\log p_0(\mathbf{Z}; \boldsymbol{\theta}_s)$ will require invoking the ODE solver in each intermediate time step [38, 39]. Such complexity is prohibitively costly for online neural applications. To circumvent this issue, we can reform Eq. (9) as follows:

$$-\mathbb{E}_{\mathbf{Z} \sim q(\mathbf{Z} | \mathbf{X}^{(t)}; \boldsymbol{\phi})} \left[ \log p_0(\mathbf{Z}; \boldsymbol{\theta}_s) \right] = \mathbb{D}_{\mathrm{KL}} \left( q(\mathbf{Z} | \mathbf{X}^{(t)}; \boldsymbol{\phi}) \| p_0(\mathbf{Z}; \boldsymbol{\theta}_s) \right) + \mathbb{H} \left( q(\mathbf{Z} | \mathbf{X}^{(t)}; \boldsymbol{\phi}) \right), \quad (10)$$

where the first term is the KL divergence from the DM marginal distribution $p_0(\mathbf{Z}; \boldsymbol{\theta}_s)$ to the probabilistic encoder distribution $q(\mathbf{Z} | \mathbf{X}^{(t)}; \boldsymbol{\phi})$, and the second term $\mathbb{H}(\cdot)$ denotes the differential entropy. For the $\mathbb{D}_{\mathrm{KL}}(\cdot)$ term in Eq. (10), via the Girsanov theorem [40, 41], we have

$$\mathbb{D}_{\mathrm{KL}} \left( q(\mathbf{Z} | \mathbf{X}^{(t)}; \boldsymbol{\phi}) \| p_0(\mathbf{Z}; \boldsymbol{\theta}_s) \right) \leqslant \mathcal{L}_{\mathrm{DSM}}(\boldsymbol{\phi}, \boldsymbol{\theta}_s) + \mathbb{D}_{\mathrm{KL}} \left( p_T(\mathbf{Z}; \boldsymbol{\theta}_s) \| \pi(\mathbf{Z}) \right), \quad (11)$$

where $\mathcal{L}_{\mathrm{DSM}}$ is the denoising score matching loss in Eq. (5), and $p_T(\cdot)$ is the distribution at final time step $T$ of Eq. (2). Consequently, we could obtain an upper bound of the maximum likelihood objective, as follows (we provide detailed derivation in Appendix B):

$$-\mathbb{E}_{\mathbf{Z} \sim q(\mathbf{Z} | \mathbf{X}^{(t)}; \boldsymbol{\phi})} \left[ \log p_0(\mathbf{Z}; \boldsymbol{\theta}_s) \right] \leqslant \underbrace{\mathbb{D}_{\mathrm{KL}} \left( p_T(\mathbf{Z}; \boldsymbol{\theta}_s) \| \pi(\mathbf{Z}) \right)}_{\text{Constant Term}}$$

$$+ \mathbb{E}_{t \sim \mathcal{U}[0,T]} \mathbb{E}_{\mathbf{Z}_0 \sim q(\mathbf{Z} | \mathbf{X}^{(t)}; \boldsymbol{\phi}), \epsilon \sim \mathcal{N}(0, \boldsymbol{I}_{l \times d})} \left[ \underbrace{w(t)^2 \left\| \epsilon - \epsilon(\mathbf{Z}_t, t; \boldsymbol{\theta}_s) \right\|_2^2}_{\text{Weighted Noise Residual}} - \underbrace{2\nabla_{\mathbf{Z}} \cdot \boldsymbol{f}(\mathbf{Z}_t, t)}_{\text{Divergence}} \right]. \quad (12)$$

Since $\pi(\mathbf{Z})$ is a fixed prior distribution, it does not depend on parameter $\boldsymbol{\phi}$. Thus, our optimization objective will include only the latter two terms, which are more computationally tractable. The first objective simplifies to a weighted noise residual for the parameter set $\boldsymbol{\phi}$ and the second divergence objective can be approximated using the Hutchinson-Skilling trace estimator [42]. We note that the recovery of spatio-temporal structure is primarily conducted by the weighted noise residual part, in which the probabilistic encoder obtains alignment guidance in awareness of the spatio-temporal structure from $\epsilon(\mathbf{Z}, t; \boldsymbol{\theta}_s)$. This procedure is illustrated in Figure 2(B).

In distribution alignment, it is a common practice to directly leverage the ground-truth data samples by introducing a regularizer term in the optimization function. To encourage the diversity of latent dynamics after alignment, here we further compute and penalize the Sinkhorn Divergence [43] between the latent dynamics samples of source domain $\mathbf{Z}^{(s)} \sim q(\cdot \mid \mathbf{X}^{(s)}; \boldsymbol{\phi}_s)$ and that of target domain $\mathbf{Z}^{(t)} \sim q(\cdot \mid \mathbf{X}^{(t)}; \boldsymbol{\phi})$:

$$\min_{\gamma} \langle \boldsymbol{\gamma}, \mathbf{C} \rangle_F + \lambda \mathbb{H}(\boldsymbol{\gamma}), \tag{13}$$

where each value $\mathbf{C}[i][j] = \left\| \mathbf{Z}_i^{(s)} - \mathbf{Z}_j^{(t)} \right\|_2^2$ in matrix $\mathbf{C}$ denotes the squared Euclidean cost to move a probability mass from $\mathbf{Z}_i^{(s)}$ to $\mathbf{Z}_j^{(t)}$, and $\mathbb{H}(\boldsymbol{\gamma})$ computes the entropy of transport plan $\boldsymbol{\gamma}$. The total loss for distribution alignment is composed of the term in (13) and the latter two terms on the right side of (12). We note that the total loss is minimized only with respect to the probabilistic encoder parameter set $\boldsymbol{\phi}$. (see Appendix C for the total loss formula and the detailed algorithm of ERDiff.)

## 4    Experiments

**Datasets.**  We first train and evaluate ERDiff with a synthetic dataset. Then we apply ERDiff to a non-human primate (NHP) dataset with neural recordings from the primary motor cortex (M1), in which the primates are performing a center-out reaching task in 8 different directions. The NHP dataset contains rich cross-day and inter-subject settings that provide us with an ideal test bed.

**Baselines for comparison.**  We compare ERDiff against the following two strong baselines proposed for the neural distribution alignment task:
- **JSDM** [24]: a metric-based method that leverages discrete samples from both the source and target domains. The alignment is performed through the symmetric Jensen–Shannon divergence [44].
- **Cycle-GAN** [21]: a state-of-the-art GAN-based method that uses cycle-consistent adversarial networks to align the distributions of latent dynamics.

Considering the neural observations and latent dynamics are in the format of multi-variate time series, we also compare ERDiff with the following methods aiming at distribution alignment for general time series data:
- **SASA** [45]: a metric-based distribution alignment method for time series data regression task through the extraction of domain-invariant representation.
- **DANN** [46]: an adversarial learning framework in which a domain classifier is followed by a feature extractor through a gradient reversal layer. This layer adjusts the gradient by multiplying it with a predefined negative constant during the training process.
- **RDA-MMD** [47]: a distribution alignment method via minimizing MMD Loss between the latent dynamics extracted from LSTM.
- **DAF** [48]: an adversarial learning framework that uses a transformer-based shared module with a domain discriminator. During the adaptation step, the domain-invariant features are invariant ($\mathbf{Q}$, $\mathbf{K}$ of self-attention); the domain-specific features ($\mathbf{V}$ of self-attention) keep tuning.

### 4.1    Synthetic dataset

**Data synthesis and evaluation metrics.**  We first generate a simulated latent dynamics dataset to illustrate the effect of our ERDiff method on spatio-temporal structure-preserving and distribution alignment performance. In this setting, we consider modeling the nonlinear latent dynamics to follow conditionally Continuous Bernoulli (CB) [49] distribution. For each single-trial latent dynamics, we generate 2-dimensional latent variables $\mathbf{Z} = \{\mathbf{z}_{1:L}\}$ and their 32-dimensional observations $\mathbf{X} = \{\mathbf{x}_{1:L}\}$, where $L = 32$. We use the following synthesis process and parameter settings to generate samples for the source and target domains, respectively:

$$p\left(\mathbf{z}_{l+1}^{(s)} \mid \mathbf{z}_l^{(s)}\right) = \prod_d \mathcal{CB}\left(\mathbf{z}_{l+1,d}^{(s)} \mid \mathbf{W}^{(s)} \tanh(\mathbf{z}_{l,d}^{(s)})\right), \quad p\left(\mathbf{x}_l^{(s)} \mid \mathbf{z}_l^{(s)}\right) = \mathcal{N}\left(\mathbf{x}_l^{(s)} \mid \mathbf{R}^{(s)} \mathbf{z}_l^{(s)}, \mathbf{K}\right),$$

$$p\left(\mathbf{z}_{l+1}^{(t)} \mid \mathbf{z}_l^{(t)}\right) = \prod_d \mathcal{CB}\left(\mathbf{z}_{l+1,d}^{(t)} \mid \mathbf{W}^{(t)} \tanh(\mathbf{z}_{l,d}^{(t)})\right), \quad p\left(\mathbf{x}_l^{(t)} \mid \mathbf{z}_l^{(t)}\right) = \mathcal{N}\left(\mathbf{x}_l^{(t)} \mid \mathbf{R}^{(t)} \mathbf{z}_l^{(t)}, \mathbf{K}\right),$$
$$\tag{14}$$

where $l \in \{1, \ldots, L\}$, and $\{\mathbf{W}^{(s)}, \mathbf{R}^{(s)}\}$, $\{\mathbf{W}^{(t)}, \mathbf{R}^{(t)}\}$ are the specific parameter sets of the source and target domains. To compare and evaluate the latent dynamics alignment performance, we estimate the trial-average log density of the aligned latent dynamics evaluated at the optimal generation

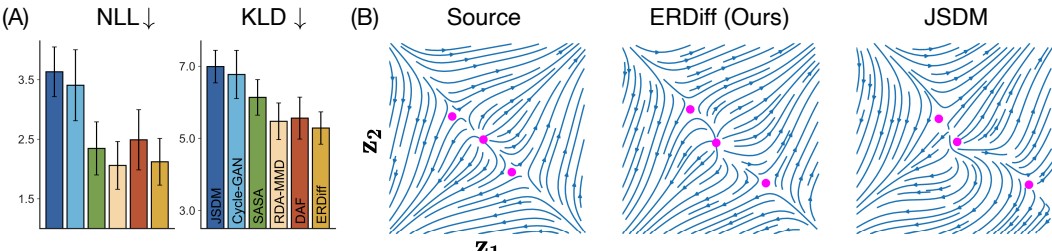

Figure 3: **Experimental results on the synthetic dataset**. **(A)** Performance comparison on trial-average negative log-likelihood (NLL) and KL Divergence (KLD). ↓ means the lower the better. ERDiff achieves the second-lowest NLL and the lowest KLD. **(B)** True continuous Bernoulli dynamics in the source domain compared to the latent dynamics aligned by ERDiff and JSDM in the target domain (blue dots denote the fixed points). ERDiff preserves the spatio-temporal structure of latent dynamics much better.

distribution: $1/L \sum_{l=0}^{L-1} \log q^* \left( \mathbf{z}_l^{(t)} \right)$, and the trial-averaged KL Divergence to the optimal latent dynamics distribution: $1/L \sum_{l=0}^{L-1} \mathbb{D}_{\mathrm{KL}} \left( p_{\phi^*}(\mathbf{z}_{l+1}^{(t)} \mid \mathbf{z}_l^{(t)}) \| p_{\phi^{(t)}}(\mathbf{z}_{l+1}^{(t)} \mid \mathbf{z}_l^{(t)}) \right)$.

**Results on synthetic dataset.** We repeat the simulation experiment five times and report the mean and standard deviation of each method in the above two quantitative evaluation metrics, shown in Figure 3(A). We observe that ERDiff achieves higher alignment performance on both two evaluation metrics compared to baseline methods. For further analysis, we plot the phase portrait of the true source domain and those inferred by ERDiff and JSDM in Figure 3(B). Compared to JSDM, ERDiff can extract and recover the spatio-temporal structure of the synthetic latent dynamics more precisely and be much closer to the ground truth. These results mainly due to the fact that ERDiff obtains structure-aware alignment signals from the DM while JSDM neglects this structural information.

## 4.2 Neural datasets

We conduct extensive experiments on two real-world neural datasets: the non-human-primate (NHP) primary motor cortex (M1) [8] and Rat hippocampal CA1 [27]. Experiments on these two datasets use a nearly identical setup. Here we primarily discuss the experimental approach and results related to the NHP motor cortex dataset. (See Appendix D for the detailed results on the rat hippocampal CA1 dataset.)

**Motor cortex dataset description.** We conduct experiments with datasets collected from the primary motor cortex (M1) of two non-human primates ('C' & 'M') [8]. The primates have been trained to reach one of eight targets at different angles (Figure 4A). Neural recordings from these two primates have been widely studied [20, 50]. During such a process, their neural spike activities (signals) in the primary motor cortex (M1) along with the reaching behavior velocity were recorded. They performed the center-out reaching task multiple times in each direction and only successful trials were saved. For our experimental evaluation purpose, we select the trials from three recording sessions for each primate per day. In total, we have 3 days for each primate. We will perform **cross-day** (recordings of the same primate performing the task on different days) and **inter-subject** (recordings of different primates) experiments.

**Data processing and evaluation metrics.** The neural recordings of each day and each primate consist of about 180-220 trials across 3 sessions. For each trial, about 200 neurons are recorded and the number of time bins is 39 with 20ms intervals. We also bin the velocity of the primate's behavior into 39 bins. Therefore, we have time-aligned neural data and behavioral data. When training with the source data, we optimize the VAE model together with the DM. One thing we need to emphasize here is that we also include a velocity-decoding loss to the VAE loss. The decoder maps the neural latent to the velocity values, which is a ridge regression model. Therefore, the inferred latent contains a rich amount of velocity information. During testing, we align the test neural data to the training neural data so that we can directly apply the velocity decoder to the latent in the test data without performance degradation. In the training session, the ratio of training and validation set is split as 80%:20% through 5-fold cross-validation. The post-processed dataset of primate 'C' contains 586 trials in total while that of primate 'M' contains 632 trials. For the evaluation protocol, since

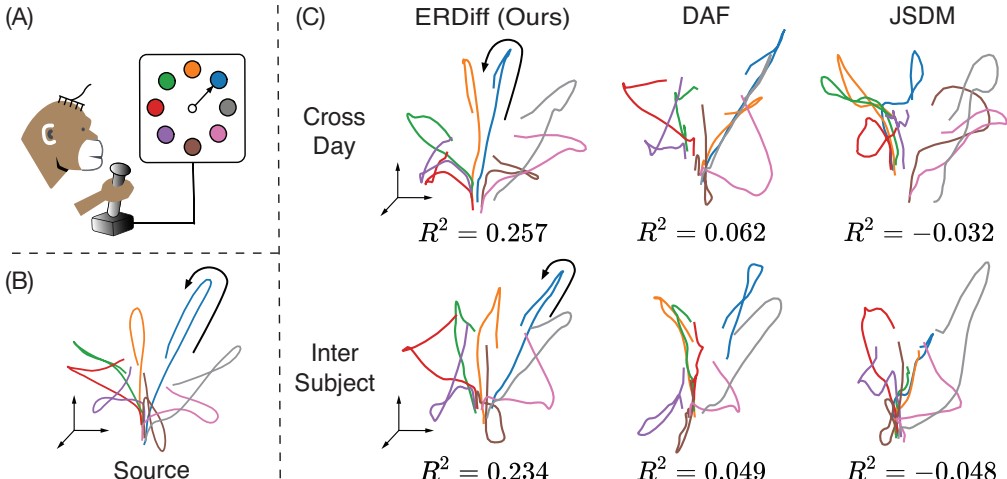

Figure 4: **Motor cortex dataset and experimental results**. **(A)** Illustration of the center-out reaching task of non-human primates. **(B)** The 3D visualization of trial-averaged latent dynamics corresponding to each reaching direction in the source domain. **(C)** The 3D visualization of trial-averaged latent dynamics corresponding to each reaching direction aligned by ERDiff, DAF, and JSDM given the target distribution from cross-day and inter-subject settings. We observe that ERDiff preserves the spatio-temporal structure of latent dynamics well.

the ground-truth source domain distribution of latent dynamics is inaccessible, we use the behavior decoding performance to evaluate the performance of latent dynamics alignment. Here we compare the true behavior velocity with the decoded one in the test data using coefficient of determination values ($R^2$, in %) and root-mean-square error (*RMSE*). To verify the generalization of each method in latent dynamics alignment, we make full use of the dataset collected in chronological order. We perform 6 sets of cross-day experiments and 10 sets of inter-subject experiments, all repeated over 5 different random seeds.

**Experimental setup.** The VAE is based on a sequential architecture [51], in which recurrent units are applied in both the probabilistic encoder and decoder. We also add domain knowledge of our alignment task into the model structure: a behavior regression decoder is cooperatively trained from the latent dynamics so that the behavior semantics information is complementarily provided during the neural manifold learning. Poisson negative log-likelihood loss is used for firing rate reconstruction and mean squared error is used for behavior regression. We use the Adam Optimizer [52] for optimization and the learning rate is chosen among $\{0.005, 0.01\}$. The batch size is uniformly set as 64. Despite the varying size of the input dimension (due to the varying number of recorded neurons in different sessions), the latent space dimension size is set as 8 for all the methods for a fair comparison. We use the dropout technique [53] and the ELU activation function [54] between layers in our probabilistic encoder and decoder architectures. During latent dynamics alignment, we perform linear probing only on the read-in layer of the probabilistic encoder while keeping the remaining layers fixed.

**Neural manifold analysis.** Considering the interpretability [50] and strong latent semantics contained in neural manifold [8], we conduct a case study based on the visualization of neural manifolds to verify the spatio-temporal structure preserving capability and alignment performance of ERDiff. In Figure 4(B), we plot the averaged latent dynamics of each direction in the source domain, which is based on one recording on primate 'C' using demixed Principle Component Analysis (dPCA) [55]. The parameters of dPCA fit with the source-domain latent dynamics while being fixed when applied to perform the transformation in the target domain. In Figure 4(C), we plot the averaged latent dynamics of each direction aligned by ERDiff and two representative baseline methods (DAF and JSDM) under both cross-day and inter-subject settings.

Under both experimental settings, the overall observation is that the spatio-temporal structure of the aligned results of ERDiff is much more coherent with that of the source domain. The results of DAF and JSDM roughly recover the direction of dynamics but fail to preserve the spatio-temporal

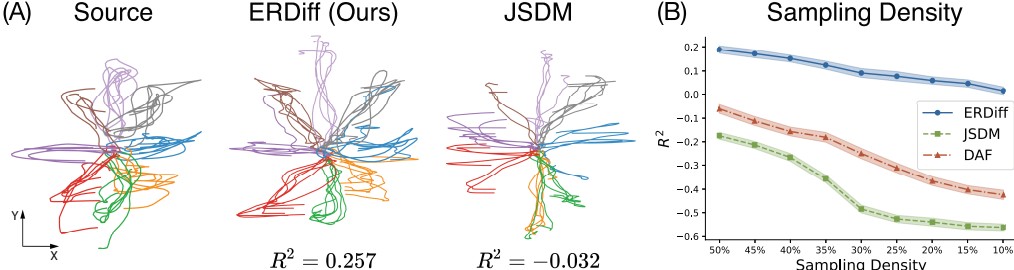

Figure 5: **(A)** True source-domain trial velocities and behavior decoding trajectories inferred from a ridge regression model given the latent dynamics aligned by ERDiff and JSDM, respectively. We can observe that ERDiff not only preserves the spatio-temporal structure but also decodes the direction more accurately. **(B)** We compare the decoding performance of ERDiff, DAF, and JSDM with a decrease in the sampling density of trials on the target domain. We can observe that ERDiff maintains a relatively high accuracy under low sampling densities.

Table 1: The R-squared values ($R^2$, in %) and RMSE of the methods on the motor cortex dataset. ERDiff w/o S is short for a variant of our proposed method that removes the spatial transformer layer in the DM. ERDiff w/o T is short for a variant of our proposed method that removes the temporal transformer layer in the DM. The boldface denotes the highest score. Each experiment condition is repeated with 5 runs, and their mean and standard deviation are listed.

| Method | Cross-Day | | Inter-Subject | |
|---|---|---|---|---|
| | $R^2(\%) \uparrow$ | RMSE $\downarrow$ | $R^2(\%) \uparrow$ | RMSE $\downarrow$ |
| Cycle-GAN | -24.83 ($\pm$3.91) | 11.28 ($\pm$0.44) | -25.47 ($\pm$3.87) | 12.23 ($\pm$0.46) |
| JSDM | -17.36 ($\pm$2.57) | 9.01 ($\pm$0.38) | -19.59 ($\pm$2.77) | 11.55 ($\pm$0.52) |
| SASA | -12.66 ($\pm$2.40) | 8.36 ($\pm$0.32) | -14.33 ($\pm$3.05) | 10.62 ($\pm$0.40) |
| DANN | -12.57 ($\pm$3.28) | 8.28 ($\pm$0.32) | -18.37 ($\pm$3.24) | 10.66 ($\pm$0.57) |
| RDA-MMD | -9.96 ($\pm$2.63) | 8.51 ($\pm$0.31) | -6.31 ($\pm$2.19) | 10.29 ($\pm$0.42) |
| DAF | -6.37 ($\pm$3.72) | 8.17 ($\pm$0.48) | -11.26 ($\pm$3.64) | **9.57**($\pm$0.58) |
| ERDiff w/o S | -12.69 ($\pm$2.64) | 8.57 ($\pm$0.50) | -14.60 ($\pm$2.88) | 10.85 ($\pm$0.57) |
| ERDiff w/o T | -14.61 ($\pm$2.33) | 8.93 ($\pm$0.50) | -17.10 ($\pm$3.23) | 10.94 ($\pm$0.59) |
| **ERDiff (Ours)** | **18.81**($\pm$2.24) | **7.99**($\pm$0.43) | **10.29**($\pm$2.86) | 9.78($\pm$0.50) |

structure tightly. That is because JSDM neglects the spatio-temporal structure during alignment and it is difficult for adversarial networks to capture such structure implicitly in DAF. Additionally, the averaged latent dynamics of each direction are much more clearly separated through ERDiff. We owe this outcome to the fact that providing extra guidance on the spatio-temporal structure would also facilitate the model to align directions properly. Additionally, without any mean offset alignment, the starting points (from the bottom center) and ending points of latent dynamics are also aligned well with the source domain, further verifying the structure recovering ability of ERDiff.

**Decoding performance comparison.** Table 1 shows a comparison of the r-squared value ($R^2$) and average RMSE on both the cross-day and inter-subject settings. While Figure 5(A) depicted the decoded velocity trajectories of a subset of trials on the cross-day setting given the method. We have the following observations: (1) Compared to traditional alignment methods specially designed for neural data, deep learning-based methods additionally model the sequential information of the latent dynamics, thus achieving better alignment results, which reflects the importance of spatio-temporal structure modeling. In most cases, ERDiff achieves the highest decoding accuracy and alignment performance among all methods. (2) From the ablation study shown at the bottom of Table 1, we find that both the Spatial Transformer layer and Temporal Transformer layer are key components in ERDiff, verifying the effectiveness of spatio-temporal structure modeling. (3) As shown in Figure 5(A), the spatio-temporal structure of the latent dynamics is well-preserved in the result of ERDiff. Compared to baselines, the smoothness and continuity of the trajectory decoded by ERDiff are also more satisfying.

**Impact of sampling density in the target domain.** We verify the consistent performance of ERDiff in few-shot target-sample circumstances. In Figure 5(B), we analyze the impact of the sampling density of the target domain on decoding performance. The setting is that we sample a portion

of target-domain data to learn the alignment and apply the alignment to the entire target domain. Despite the sampling density drops from $50\%$ to $10\%$, our results demonstrate that ERDiff continues to produce fairly consistent decoding accuracy with a small drop. This result validates our argument that ERDiff both preserves the dynamics structure underlying neural activities and maintains the characteristics of the target domain. In comparison, the performance of baseline methods shrinks drastically because they lack prior knowledge of the spatio-temporal structure.

We can conclude that the DM in ERDiff is capable of extracting the spatio-temporal structure in the source domain latent dynamics, providing a valuable inductive bias in recovering such structure during distribution alignment.

## 5 Discussion

In this work, we propose a new method named ERDiff, for solving the neural distribution alignment issue in real-world neuroscience applications (e.g., brain-computer interfaces). Firstly, with the source domain, we propose to use a diffusion model to extract the spatio-temporal structure within the latent dynamics of trials. Next, in the alignment phase with the target domain, the spatio-temporal structure of latent dynamics is recovered through the maximum likelihood alignment based on the diffusion model. Experimental results on synthetic and real-world motor cortex datasets verify the effectiveness of ERDiff in the enhancement of long-term robustness and behavior decoding performance from neural latent dynamics.

To be in line with the conventions of previous studies on neural distribution alignment [4, 16], the behavioral (velocity) signals of the source domain are present during the VAE training. These signals do help in learning a more interpretable neural latent space. However, we emphasize that ERDiff does not incorporate any behavioral signals of the target domain during the distribution alignment phase. Hence, ERDiff is entirely an *unsupervised* neural distribution alignment (i.e., test-time adaptation) method. As for the computational cost analysis, In the source domain training phase, the additional computation cost of ERDiff primarily comes from the diffusion model (DM) training. We note that the DM is trained in the latent space $\mathbf{Z}$, which is significantly lower in dimensionality than the raw neural spiking signal space $\mathbf{X}$. Therefore, the computational overhead of this one-time training phase is acceptable, especially given that it can be conducted offline in real-world BCI applications. In the alignment phase, we would like to emphasize that ERDiff maintains a comparable computational cost with baseline methods. Please refer to Appendix E for a comprehensive analysis of ERDiff's computational cost and time complexity.

**Limitation and future work.** (1) Currently, ERDiff can align well with a single source domain neural latent distribution. An intriguing direction for future work would be learning a unified latent space across multiple source domains using the diffusion model. Thus the method would be applicable to domain generalization problems. (2) Generalization on alternative latent variable models (LVM). In this paper, ERDiff identifies the latent variables of raw neural spiking signals with a canonical version of VAE. However, the architecture of the LVM within ERDiff is actually disentangled from the diffusion model training or MLA procedure. Future work includes validating ERDiff given more advanced implementations of LVM, e.g., LFADS [56], STNDT [32].

**Broader impact.** Not confined to practical applications in systems neuroscience, the maximum likelihood alignment (MLA) with diffusion model algorithm proposed in ERDiff has great potential to apply to broader domain adaptation tasks across general time-series datasets (e.g., weather forecasting, and seismology). We also expect that our method can be applied or extended to other real-world scenarios and the broader field of neuroscience/AI.

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

## Appendix to "Extraction and Recovery of Spatio-Temporal Structure in Latent Dynamics Alignment with Diffusion Model"

## A  Methodology details

### A.1  DM architecture details

We adopt the architecture of DM mainly derived from Diffusion Transformer (DiT) [36]. The canonical DiT architecture is based on techniques like patchify [57] and transformer layer for tokens [58], which are well-suited for image feature extraction. This is because the above techniques focus on local feature extraction and the global feature can also be implicitly captured through the stacking of token-based Transformer layers. However, considering the neural observations and latent dynamics are in the format of multi-variate time series, patchify and local feature extraction loses their semantic meaning. There doesn't exist a theoretical guarantee or bio-plausible observation that adjacent latent dimensions of the data matrix have a stronger connection than far-apart latent dimensions. Thus, directly adopting the traditional DiT architecture into this setting may lead to sub-optimal solutions.

To fully utilize the domain knowledge of our task, we propose to use the Spatio-Temporal Transformer Block (STBlock). Each STBlock is mainly composed of a Spatio Transformer layer followed by a Temporal Transformer layer, which are 1-layer encoders based on multi-head self-attention. Since there exists underlying dependency and structure between latent state dimensions, the Spatio Transformer layer takes latent states of each time bin as inputs to extract their spatial structure. Whereas the Temporal Transformer layer takes the entire latent trajectory of each latent space dimension as inputs to extract its underlying temporal structure. We note that we use the sinusoidal position embeddings [59] to encode the timestep $t$ (i.e., noise scale) into the deep neural network of DM. In each STBlock, the input sequentially goes through:

- **Spatio Transformer**: Layer Normalization $\rightarrow$ Multi-head Self-attention Layer (along time bins) $\rightarrow$ Point-wise Feed-forward.

- **Temporal Transformer**: Layer Normalization $\rightarrow$ Multi-head Self-attention Layer (along latent space dimensions) $\rightarrow$ Point-wise Feed-forward.

We illustrate the main architecture of DM in Figure 2(A), and implement the DM in Pytorch [60].

### A.2  VAE and DM cooperative source domain learning details

In DM training, we note that $\mathbf{Z}_0^{(s)}$ here are actually latent dynamics inferred via VAE in Eq. 6. Considering the limited number of source-domain latent dynamics, we wish to perform data augmentation so that the DM can adequately estimate $p_s(\mathbf{Z})$. Here we propose to enrich the input samples by learning the VAE objective (Eq. 6) and the diffusion objective (Eq. 8) cooperatively. Through the learning process of the VAE objective (i.e., ELBO), the optimization process with stochastic gradient descent (SGD) adds auxiliary perturbation to the original data samples $\mathbf{Z}_0^{(s)}$ rather than pure Gaussian noise. This technique further fills the sample space of $\mathbf{Z}_0^{(s)}$, leading to better density estimation. Specifically, in each training iteration, conditioning on the current value of $\phi_s$, we infer a set of $\mathbf{Z}_0 = h(\mathbf{X}^{(s)}; \phi_s)$ and use it as the temporal $\mathbf{Z}_0^{(s)}$ to optimize Eq. 8. The traditionally sequential approach is that we fully

Table 2: The coefficient of determination values ($R^2 \uparrow$, in %) and RMSE $\downarrow$ of sequential source domain learning and cooperative source domain learning on the primate motor cortex dataset. Boldface denotes the highest score. Each experiment condition is repeated with 5 different random seeds, and their mean and standard deviation are listed.

|        | Metric       | Sequential         | Cooperative              |
|--------|--------------|--------------------|--------------------------|
| M1-M2  | $R^2(\%)$    | 18.96 ($\pm$2.27)  | **20.47** ($\pm$2.71)    |
|        | RMSE         | 7.76 ($\pm$0.40)   | **7.62** ($\pm$0.42)     |
| M2-M3  | $R^2(\%)$    | 21.73 ($\pm$2.46)  | **22.62** ($\pm$2.66)    |
|        | RMSE         | 7.94 ($\pm$0.47)   | **7.73** ($\pm$0.50)     |
| M2-C2  | $R^2(\%)$    | 6.96 ($\pm$2.88)   | **8.57** ($\pm$2.96)     |
|        | RMSE         | 11.85 ($\pm$0.42)  | **11.64** ($\pm$0.51)    |

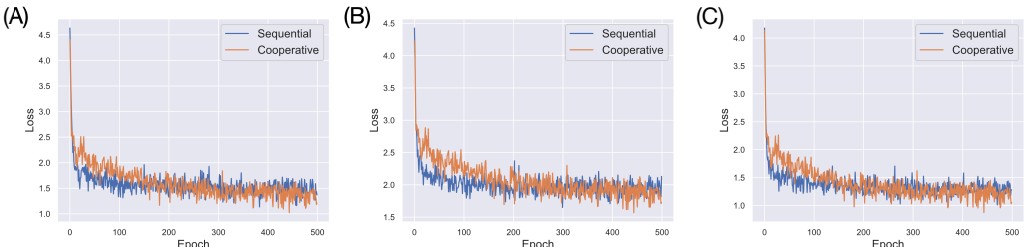

Figure 6: **Training loss curves under different neural sessions**. **(A)** Neural session: M-1. **(B)** Neural session: M-3. **(C)** Neural session: C-2.

optimize VAE first, obtain an optimal $\phi_s$, and use it to optimize Eq. 8. Experimental results show that the former approach achieves higher density estimation and alignment performance. Figure 6 manifests the training loss curve in three source-target neural recording session pairs. We observe that despite the relatively under-fitting at the early stage, the cooperative source domain learning paradigm converges to solutions with lower losses and better fits. Table 2 manifests that our cooperative source domain learning paradigm leads to higher distribution alignment and neural decoding performance.

## B    Detailed derivation of maximum likelihood alignment

### B.1    Relationship between KL-Divergence and DSM Loss

Under assumptions in Appendix A of [41], the KL divergence between the ground truth density $q\left(\mathbf{Z}\mid\mathbf{X}^{(t)};\phi\right)$ and the DM marginal distribution $p_0\left(\mathbf{Z};\boldsymbol{\theta}_s\right)$ can be derived as:

$$
\begin{aligned}
&\mathbb{D}_{\mathrm{KL}}\left(q(\mathbf{Z}\mid\mathbf{X}^{(t)};\phi)\|p_0\left(\mathbf{Z};\boldsymbol{\theta}_s\right)\right)\\
&\overset{(i)}{\leqslant} \mathbb{E}_{t\sim\mathcal{U}[0,T]}\mathbb{E}_{\mathbf{Z}\sim q(\cdot|\mathbf{X}^{(t)};\phi)}\left[\lambda(t)\left(\nabla_{\mathbf{Z}}\log p_t(\mathbf{Z};\boldsymbol{\theta}_s)-\boldsymbol{s}(\mathbf{Z},t;\boldsymbol{\theta}_s)\right)\mathrm{d}\overline{\mathbf{w}}\right]+\mathbb{D}_{\mathrm{KL}}\left(p_T\left(\mathbf{Z};\boldsymbol{\theta}_s\right)\|\pi(\mathbf{Z})\right)\\
&\quad+\frac{1}{2}\mathbb{E}_{t\sim\mathcal{U}[0,T]}\mathbb{E}_{\mathbf{Z}\sim q(\cdot|\mathbf{X}^{(t)};\phi)}\left[\lambda(t)^2\left\|\nabla_{\mathbf{Z}}\log p_t(\mathbf{Z};\boldsymbol{\theta}_s)-\boldsymbol{s}(\mathbf{Z},t;\boldsymbol{\theta}_s)\right\|_2^2\,\mathrm{d}t\right]\\
&\overset{(ii)}{=} \mathbb{E}_{t\sim\mathcal{U}[0,T]}\mathbb{E}_{\mathbf{Z}\sim q(\cdot|\mathbf{X}^{(t)};\phi)}\left[\lambda(t)^2\left\|\nabla_{\mathbf{Z}}\log p_t(\mathbf{Z};\boldsymbol{\theta}_s)-\boldsymbol{s}(\mathbf{Z},t;\boldsymbol{\theta}_s)\right\|_2^2\,\mathrm{d}t\right]+\mathbb{D}_{\mathrm{KL}}\left(p_T\left(\mathbf{Z};\boldsymbol{\theta}_s\right)\|\pi(\mathbf{Z})\right)\\
&\overset{(iii)}{=} \mathbb{E}_{t\sim\mathcal{U}[0,T]}\mathbb{E}_{\mathbf{Z}_0\sim q(\cdot|\mathbf{X}^{(t)};\phi),p_{0t}(\mathbf{Z}_t|\mathbf{Z}_0)}\left[\lambda(t)^2\left\|\nabla_{\mathbf{Z}_t}\log p_{0t}\left(\mathbf{Z}_t\mid\mathbf{Z}_0\right)-\boldsymbol{s}\left(\mathbf{Z}_t,t;\boldsymbol{\theta}\right)\right\|_2^2\right]\\
&\quad+\mathbb{D}_{\mathrm{KL}}\left(p_T\left(\mathbf{Z};\boldsymbol{\theta}_s\right)\|\pi(\mathbf{Z})\right)\\
&=\ \mathcal{L}_{\mathrm{DSM}}\left(\phi,\boldsymbol{\theta}_s\right)+\mathbb{D}_{\mathrm{KL}}\left(p_T(\mathbf{Z};\boldsymbol{\theta}_s)\|\pi(\mathbf{Z})\right),
\end{aligned}
$$
(15)

in which $(i)$ is due to Girsanov Theorem [40], in $(ii)$ we invoke the martingale property of Itô integrals [61], and in $(iii)$ we use the denoising score matching (DSM) technique [29]. Thus we can draw to Eq. 11.

### B.2    Upper bound of maximum likelihood alignment objective

In Section 3.3, by substituting Eq. 11 into Eq. 10, we have

$$
-\mathbb{E}_{\mathbf{Z}\sim q(\mathbf{Z}|\mathbf{X}^{(t)};\phi)}\left[\log p_0(\mathbf{Z};\boldsymbol{\theta}_s)\right]\leqslant\mathcal{L}_{\mathrm{DSM}}\left(\phi,\boldsymbol{\theta}_s\right)+\mathbb{D}_{\mathrm{KL}}\left(p_T(\mathbf{Z};\boldsymbol{\theta}_s)\|\pi(\mathbf{Z})\right)+\mathbb{H}\left(q(\mathbf{Z}\mid\mathbf{X}^{(t)};\phi)\right).
$$
(16)

We note that the third term $\mathbb{H}(\cdot)$ depends on the parameter set $\phi$ of the probabilistic encoder. As $q(\mathbf{Z} \mid \mathbf{X}^{(t)}; \phi) \approx p_0(\mathbf{Z}; \boldsymbol{\theta}_s)$, we have

$$\mathbb{H}\left(q(\mathbf{Z} \mid \mathbf{X}^{(t)}; \phi)\right) - \mathbb{H}\left(p_T(\mathbf{Z}; \boldsymbol{\theta}_s)\right) \approx \int_T^0 \frac{\partial}{\partial t} \mathbb{H}\left(p_t(\mathbf{Z}; \boldsymbol{\theta}_s)\right) \mathrm{d}t \tag{17}$$

$$\overset{(i)}{=} \mathbb{E}_{t \sim \mathcal{U}[0,T]} \mathbb{E}_{\mathbf{Z} \sim q(\mathbf{Z} \mid \mathbf{X}^{(t)}; \phi)} \left[2\boldsymbol{f}(\mathbf{Z},t)^\top \nabla_\mathbf{Z} \log p_t(\mathbf{Z}; \boldsymbol{\theta}_s) - \lambda(t)^2 \|\nabla_\mathbf{Z} \log p_t(\mathbf{Z}; \boldsymbol{\theta}_s)\|_2^2\right] \mathrm{d}t$$

$$\overset{(ii)}{=} -\mathbb{E}_{t \sim \mathcal{U}[0,T]} \mathbb{E}_{\mathbf{Z} \sim q(\mathbf{Z} \mid \mathbf{X}^{(t)}; \phi)} \left[2\nabla_\mathbf{Z} \cdot \boldsymbol{f}(\mathbf{Z},t) + \lambda(t)^2 \|\nabla_\mathbf{Z} \log p_t(\mathbf{Z}; \boldsymbol{\theta}_s)\|_2^2\right] \mathrm{d}t$$

$$\overset{(iii)}{=} -\mathbb{E}_{t \sim \mathcal{U}[0,T]} \mathbb{E}_{\mathbf{Z}_0 \sim q(\cdot \mid \mathbf{X}^{(t)}; \phi), p_{0t}(\mathbf{Z}_t \mid \mathbf{Z}_0)} \left[2\nabla_{\mathbf{Z}_t} \cdot \boldsymbol{f}(\mathbf{Z}_t,t) + \lambda(t)^2 \|\nabla_{\mathbf{Z}_t} \log p_{0t}(\mathbf{Z}_t \mid \mathbf{Z}_0)\|_2^2\right] \mathrm{d}t \tag{18}$$

where in both (i) and (ii) we use integration by parts, and in (iii) we use denoising score matching (DSM) [29]. Putting the second term on the LHS of Eq. 17 into RHS and then substituting the third term on the RHS of Eq. 16, we have

$$-\mathbb{E}_{\mathbf{Z} \sim q(\mathbf{Z} \mid \mathbf{X}^{(t)}; \phi)} \left[\log p_0(\mathbf{Z}; \boldsymbol{\theta}_s)\right] \leqslant \mathbb{D}_{\mathrm{KL}}\left(p_T(\mathbf{Z}; \boldsymbol{\theta}_s)\|\pi(\mathbf{Z})\right) \tag{19}$$

$$-\mathbb{E}_{t \sim \mathcal{U}[0,T]} \mathbb{E}_{\mathbf{Z}_0 \sim q(\cdot \mid \mathbf{X}^{(t)}; \phi), p_{0t}(\mathbf{Z}_t \mid \mathbf{Z}_0)} \left[\lambda(t)^2 \|\nabla_{\mathbf{Z}_t} \log p_{0t}(\mathbf{Z}_t \mid \mathbf{Z}_0)\|_2^2\right] \tag{20}$$

$$+\mathbb{E}_{t \sim \mathcal{U}[0,T]} \mathbb{E}_{\mathbf{Z}_0 \sim q(\cdot \mid \mathbf{X}^{(t)}; \phi), p_{0t}(\mathbf{Z}_t \mid \mathbf{Z}_0)} \left[\lambda(t)^2 \|\nabla_{\mathbf{Z}_t} \log p_{0t}(\mathbf{Z}_t \mid \mathbf{Z}_0) - \boldsymbol{s}(\mathbf{Z}_t, t; \boldsymbol{\theta})\|_2^2\right] \tag{21}$$

$$-\mathbb{E}_{t \sim \mathcal{U}[0,T]} \mathbb{E}_{\mathbf{Z}_0 \sim q(\cdot \mid \mathbf{X}^{(t)}; \phi), p_{0t}(\mathbf{Z}_t \mid \mathbf{Z}_0)} \left[-2\nabla_{\mathbf{Z}_t} \cdot \boldsymbol{f}(\mathbf{Z}_t, t)\right]. \tag{22}$$

Since the transition probability $p_{0t}(\mathbf{Z}_t \mid \mathbf{Z}_0)$ is a fixed Gaussian distribution and it is independent of the parameter set $\phi$, we can eliminate the term in Eq. 20 and rewrite the above Eqs as:

$$-\mathbb{E}_{\mathbf{Z} \sim q(\mathbf{Z} \mid \mathbf{X}^{(t)}; \phi)} \left[\log p_0(\mathbf{Z}; \boldsymbol{\theta}_s)\right] \leqslant \mathbb{D}_{\mathrm{KL}}\left(p_T(\mathbf{Z}; \boldsymbol{\theta}_s)\|\pi(\mathbf{Z})\right) \tag{23}$$

$$+\mathbb{E}_{t \sim \mathcal{U}[0,T]} \mathbb{E}_{\mathbf{Z}_0 \sim q(\cdot \mid \mathbf{X}^{(t)}; \phi), p_{0t}(\mathbf{Z}_t \mid \mathbf{Z}_0)} \left[\lambda(t)^2 \|\nabla_{\mathbf{Z}_t} \log p_{0t}(\mathbf{Z}_t \mid \mathbf{Z}_0) - \boldsymbol{s}(\mathbf{Z}_t, t; \boldsymbol{\theta})\|_2^2\right] \tag{24}$$

$$-\mathbb{E}_{t \sim \mathcal{U}[0,T]} \mathbb{E}_{\mathbf{Z}_0 \sim q(\cdot \mid \mathbf{X}^{(t)}; \phi), p_{0t}(\mathbf{Z}_t \mid \mathbf{Z}_0)} \left[-2\nabla_{\mathbf{Z}_t} \cdot \boldsymbol{f}(\mathbf{Z}_t, t)\right]. \tag{25}$$

By substituting denoiser function $\boldsymbol{\epsilon}(\mathbf{Z}_t, t; \boldsymbol{\theta})$ into score function $\boldsymbol{s}(\mathbf{Z}_t, t; \boldsymbol{\theta})$ of Eq. 24, we have:

$$-\mathbb{E}_{\mathbf{Z} \sim q(\mathbf{Z} \mid \mathbf{X}^{(t)}; \phi)} \left[\log p_0(\mathbf{Z}; \boldsymbol{\theta}_s)\right] \leqslant \mathbb{D}_{\mathrm{KL}}\left(p_T(\mathbf{Z}; \boldsymbol{\theta}_s)\|\pi(\mathbf{Z})\right)$$

$$+\mathbb{E}_{t \sim \mathcal{U}[0,T]} \mathbb{E}_{\mathbf{Z}_0 \sim q(\mathbf{Z} \mid \mathbf{X}^{(t)}; \phi), \boldsymbol{\epsilon} \sim \mathcal{N}(0, \boldsymbol{I}_{l \times d})} \left[w(t)^2 \|\boldsymbol{\epsilon} - \boldsymbol{\epsilon}(\mathbf{Z}_t, t; \boldsymbol{\theta}_s)\|_2^2 - 2\nabla_{\mathbf{Z}_t} \cdot \boldsymbol{f}(\mathbf{Z}_t, t)\right].$$

## C Detailed algorithm

### C.1 Overall alignment loss function in ERDiff

Extracting the latter two terms from Eq. 12, we have the following main Maximum Likelihood Alignment (MLA) loss:

$$\mathcal{L}_{\mathrm{MLA}}(\phi) = \mathbb{E}_{t \sim \mathcal{U}[0,T]} \mathbb{E}_{\mathbf{Z}_0 \sim q(\mathbf{Z} \mid \mathbf{X}^{(t)}; \phi), \boldsymbol{\epsilon} \sim \mathcal{N}(0, \boldsymbol{I}_{l \times d})} \left[w(t)^2 \|\boldsymbol{\epsilon} - \boldsymbol{\epsilon}(\mathbf{Z}_t, t; \boldsymbol{\theta}_s)\|_2^2 - 2\nabla_{\mathbf{Z}_t} \cdot \boldsymbol{f}(\mathbf{Z}_t, t)\right]. \tag{26}$$

Then, considering the Sinkhorn Regularizer term in Eq. 13:

$$\mathcal{L}_{\mathrm{SHD}}(\phi) = \min_\gamma \langle \boldsymbol{\gamma}, \mathbf{C} \rangle_F + \lambda \mathbb{H}(\boldsymbol{\gamma}), \tag{27}$$

where each value $\mathbf{C}[i][j] = \left\|\mathbf{Z}_i^{(s)} - \mathbf{Z}_j^{(t)}\right\|_2^2$ in cost matrix $\mathbf{C}$ denotes the squared Euclidean cost from $\mathbf{Z}_i^{(s)}$ to $\mathbf{Z}_j^{(t)}$, $\mathbb{H}(\boldsymbol{\gamma})$ computes the entropy of transport plan $\boldsymbol{\gamma}$, and $\lambda$ refers to the weight of the entropy term. Then, we can find the optimal $\phi_t$ via the minimization of the following total loss function:

$$\phi_t = \underset{\phi}{\operatorname{argmin}} \left[(1-\alpha)\mathcal{L}_{\mathrm{MLA}} + \alpha\mathcal{L}_{\mathrm{SHD}}\right], \tag{28}$$

where $\alpha \in [0, 1]$ is a trade-off parameter that weights the importance of Sinkhorn Regularizer term.

## C.2 Algorithm for source domain learning of ERDiff

---
**Algorithm 1** Source Domain Learning of ERDiff

---
**Input:** Source-domain neural observations $\mathbf{X}^{(s)}$; Learning rate $\eta$ and all other hyperparameters.
**Output:** Parameter set $\phi^{(s)}$ of source-domain probabilistic encoder; Parameter set $\psi^{(s)}$ of source-domain probabilistic decoder; Parameter set $\theta^{(s)}$ of source-domain diffusion model.
 1: Initialize $\phi, \psi,$ and $\theta$;
 2: **while** not converge **do**
 3:     $\mathcal{L}_{\mathrm{ELBO}} \leftarrow$ Compute the loss function based on evidence lower bound according to Eq. 6;
 4:     Parameter Update: $\phi \leftarrow \phi - \eta \cdot \partial \mathcal{L}_{\mathrm{ELBO}}/\partial\phi$;
 5:     Parameter Update: $\psi \leftarrow \psi - \eta \cdot \partial \mathcal{L}_{\mathrm{ELBO}}/\partial\psi$;
 6:     Inference $\mathbf{Z}_0^{(s)} \sim q(\cdot \mid \mathbf{X}^{(s)}; \phi)$, and Noise Sampling $\boldsymbol{\epsilon} \sim \mathcal{N}(0, \boldsymbol{I}_{l \times d})$;
 7:     $\mathcal{L}_{\mathrm{DSM}} \leftarrow$ Compute the denoising score matching loss according to Eq. 5;
 8:     Parameter Update: $\theta \leftarrow \theta - \eta \cdot \partial\mathcal{L}_{\mathrm{DSM}}/\partial\theta$;
 9: **end while**
10: **return** $\phi, \psi,$ and $\theta$.

---

## C.3 Algorithm for maximum likelihood alignment of ERDiff

---
**Algorithm 2** Maximum Likelihood Alignment of ERDiff

---
**Input:** Target-domain neural observations $\mathbf{X}^{(t)}$; Learning rate $\eta$ and all other hyperparameters.
**Output:** Parameter set $\phi^{(t)}$ of target-domain probabilistic encoder;
 1: **while** not converge **do**
 2:     Inference $\mathbf{Z}_0^{(t)} \sim q(\cdot \mid \mathbf{X}^{(t)}; \phi)$, and Noise Sampling $\boldsymbol{\epsilon} \sim \mathcal{N}(0, \boldsymbol{I}_{l \times d})$;
 3:     $\mathcal{L}_{\mathrm{MLA}} \leftarrow$ Compute the main Maximum Likelihood Alignment Loss according to Eq. 26;
 4:     $\mathcal{L}_{\mathrm{SHD}} \leftarrow$ Compute the Sinkhorn Regularizer according to Eq. 27;
 5:     Parameter Update: $\phi \leftarrow \phi - \eta \cdot \partial\left[(1-\alpha)\mathcal{L}_{\mathrm{MLA}} + \alpha\mathcal{L}_{\mathrm{SHD}}\right]/\partial\phi$;
 6: **end while**
 7: **return** $\phi$.

---

# D Experimental results on rat hippocampal CA1 dataset

We further verify the effectiveness of ERDiff on a publicly available rat hippocampus dataset [27]. In this study, a rat navigated a 1.6m linear track, receiving rewards at both terminals (L&R) as depicted in Figure 4 (A). Concurrently, neural activity from the hippocampal CA1 region was captured, in which the neuron numbers across all recorded sessions are around 120. The neural spiking activities were binned into 25ms intervals. A single round trip by the rat from one end of the track to the opposite was categorized as one lap. We chose 5 sessions in total, and in each session, 80 laps were sampled. The $2-$dimensional neural latent manifolds can be observed in Figure 7 (B) and (C). We also list the neural decoding results of the rat's position in Table 3.

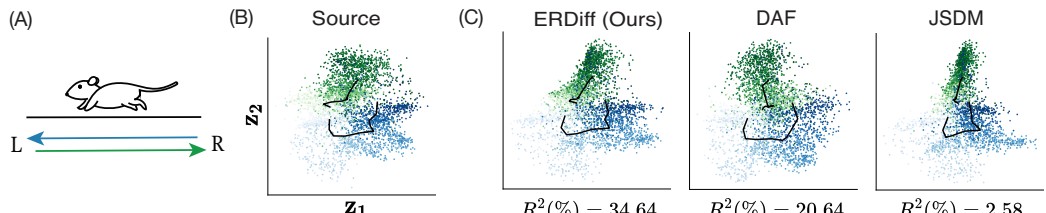

Figure 7: **Rat hippocampus dataset and experimental results**. **(A)** Illustration of the experiment setting. **(B)** 2D visualization of inferred latents in the source domain, in which the bold lines manifest the mean location of latent states corresponding to the rat's position for two opposite directions. **(C)** On target domain, 2D visualization of inferred latents after alignment by ERDiff, DAF, and JSDM, respectively. We observe that ERDiff preserves the spatio-temporal structure of latent dynamics well.

Table 3: Decoding results after neural distribution alignment on the rat hippocampus dataset across sessions. Each experiment condition is repeated with 5 different random seeds, and their mean and standard deviation are listed.

| Method | $R^2(\%) \uparrow$ | RMSE $\downarrow$ |
|---|---|---|
| Cycle-GAN | 4.81 ($\pm$2.47) | 7.65 ($\pm$0.34) |
| JSDM | -2.88 ($\pm$1.59) | 7.65 ($\pm$0.23) |
| SASA | 18.51 ($\pm$1.65) | 6.98 ($\pm$0.20) |
| DANN | 15.04 ($\pm$2.32) | 7.30 ($\pm$0.26) |
| RDA-MMD | 21.73 ($\pm$2.45) | 7.22 ($\pm$0.26) |
| DAF | 20.20 ($\pm$2.26) | 7.36 ($\pm$0.28) |
| **ERDiff** | **32.69**($\pm$2.19) | **6.84**($\pm$0.26) |

Table 4: Comparative analyses of computational cost between ERDiff and baseline methods during alignment. ERDiff has a comparable computational cost and maintains the stability of alignment.

| Method | Cycle-GAN | JSDM | SASA | RDA-MMD | DAF | **ERDiff** |
|---|---|---|---|---|---|---|
| Add'l. Param | 26K | 0K | 33K | 65K | 91K | 28K |
| Add'l. Size | 117KB | 0KB | 187KB | 314KB | 367KB | 139KB |
| Align. Time | 103ms | 77ms | 155ms | 264ms | 251ms | 183ms |
| Stability | ✗ | ✔ | ✔ | ✗ | ✗ | ✔ |

# E   Computational cost analysis

Here we conduct time complexity analysis with respect to the batch size $B$ for the alignment phase. The ERDiff's alignment objective function is composed of two main terms: Diffusion Noise Residual and Sinkhorn Divergence. We note that in the diffusion noise residual computation, it does not go through the entire $T$ diffusion steps. Instead, it just samples a single time step (noise scale) $t$ and calculates the noise residual specific to that step. Thus, the total complexity of this part takes $\mathcal{O}(K_1 * B * d)$, in which the coefficient $K_1$ relates to the inference complexity of the DM denoiser $\epsilon(\mathbf{Z}, t)$; $d$ denotes the latent dimension size. For the Sinkhorn Divergence, it has to compute the distance matrix, costing $\mathcal{O}(K_2 * B^2)$; $K_2$ is a relatively small coefficient in magnitude. By summing up, the total complexity of ERDiff is given by $\mathcal{O}(K_1 * B * d + K_2 * B^2)$. This $\mathcal{O}(B^2)$ complexity is applicable since the non-adversarial baseline methods we compared (i.e., JSDM, and SASA) require quadratic complexities as well.

For any given target domain, ERDiff can stably align it to the source domain in a comparable overhead with baselines. In Table 4, we conduct a comparative analysis between ERDiff and baseline methods in terms of additional parameter number, additional model size, stability, and alignment time. The demonstrated alignment time corresponds to the execution time for aligning one iteration (a batch of size 64) on a MacBook Pro (2019 equipped with 8-Core Intel Core i9 and 4 GB RAM).

