# OpenReview forum: "Extraction and Recovery of Spatio-Temporal Structure in Latent Dynamics Alignment with Diffusion Models"
_NeurIPS.cc/2023/Conference — NeurIPS 2023 spotlight_

### Official Review · Reviewer_1oNu · 2023-06-30

**Soundness:** 4 excellent
**Presentation:** 3 good
**Contribution:** 3 good
**Rating:** 7
**Confidence:** 3

**Summary:**

The authors address the problem of aligning behavior-related neural population dynamics, either within-subject but across different experimental sessions, or between subjects. This problem exists due to inter-subject variability in terms of which neurons are recorded or drift in the recording and is an important problem for systems neuroscience and for the development of brain-machine interfaces. Importantly, neural activity dynamics in many regions of the brain during behavior lie on a low-dimensional manifold and therefore have a defined spatio-temporal structure. While many state-of-the-art alignment approaches do not take into account this structure and therefore do not preserve it, the authors introduce a diffusion-guided method that does. The approach first uses a diffusional model to discover the manifold on which neural activity evolves (ie the spatiotemporal structure), then it uses this model to guide the alignment, which is done using MLE.

**Strengths:**

Originality + quality: The authors developed a novel approach for time-series alignment that also discovers latent structure (the low-D manifold on which the neural activity evolves).

Quality: The authors validate their model against a number of state-of-the-art alignment methods on both synthetic and real-world data, demonstrating the practical applicability of their method.

Clarity: The authors clearly state the problem and its details, as well as how their method differs from existing ones, and its advantages.

Significance: The method has significance to the alignment of timeseries with latent spatiotemporal structure, which is broadly relevant in systems neuroscience. Although the authors mainly focus on behaviorally relevant neural data, timeseries in other fields also often possess lower-dimensional latent structure, so this method could be broadly applicable. The authors approach could also be relevant to identifying latent structure outside of alignment context, although I am not familiar with how it compares to existing approaches to do so.

**Weaknesses:**

Clarity: Figure and table legends should be more clear (see Questions section)

Originality: Authors should state whether existing methods for discovering latent structures using diffusional models exist, and how their methods (e.g. architecture) differs from existing methods.

Quality: The method is validated in data with strong, low-dimensional latent structure (monkey reaching tasks). The generalizability to other types of neural dynamics time series of varying dimensionality should be evaluated to determine limitations.

**Questions:**

Figure 3B: Can the authors describe better what the takeaway from this panel is? In what ways does ERDiff improve on JSDM. The legend should tell the reader what the pink dots are (presumably critical points?).

Figure 4: Please note in the legend or figure that the R-squared value is in %. I was confused at first as to how the R-squared could be negative before reading Table 1.

Table 1: It is unclear to me what the denominator for the R-squared % calculation is. Can the authors make it more apparent in the table legend?

If the time-series do not belong on a well-defined low-dimensional manifold, will the method hallucinate something? How well does the method work for more unstructured data?

As the dimensionality of the latent structure increases, how much more badly does the method perform? The authors should quantify this.

**Limitations:**

The authors should address in more detail the kinds of latent structures/time series that would pose more of an issue for their method. Otherwise, limitations are adequately addressed.

---

> ### Author Rebuttal · Authors · 2023-08-10
>
> Thank you for your positive review and insightful questions. We provide clarifications and new results to address your questions below.
>
> > Authors should state whether existing methods for discovering latent structures using diffusional models exist.
>
> To the best of our knowledge, no existing methods use diffusion models (DM) for the discovery of neural latent structures.
>
> > The generalizability to other types of neural dynamics time series of varying dimensionality should be evaluated to determine limitations.
>
> The generalizability of each method is a vital consideration. The monkey reaching dataset has a neural signal dimension of around 180 and an average trial length of around 40. Additionally, we conducted experiments on an unconstrained rat’s hippocampus dataset [1], where the neural signal dimension is 120 and the average trial length is around 110. The corresponding results are presented in Section 1.1 of the general response.
>
> > Figure 3B: Can the authors describe better what the takeaway from this panel is? In what ways does ERDiff improve on JSDM. The legend should tell the reader what the pink dots are.
>
> We appreciate your suggestion about Fig. 3(B). In this figure, the phase portraits represent the trajectories and structures of latent dynamics. We note that, after alignment, ERDiff's phase portrait is noticeably closer to that of the source domain in comparison to JSDM. The pink dots you pointed out represent the stable fixed points within the phase portraits, which correspond to steady positions or equilibria of the dynamics system. We will provide a more detailed legend in the revised manuscript.
>
> > Table 1: It is unclear to me what the denominator for the R-squared % calculation is. Can the authors make it more apparent in the table legend?
>
> We would like to clarify that $R^2$ refers to the *coefficient of determination* that quantifies the goodness-of-fit for regression models. Formally, it is calculated as follows: $R^2(y, f) = \sum_{i=1}^{n}(y_i - \bar{y})(f_i - \bar{f})/\sum_{i=1}^{n}(y_i - \bar{y})^2$. The denominator is the total variance of ground-truth values.  We will incorporate the formula of $R^2$ calculation into the revised manuscript.
>
> >  **(1)** If the time-series do not belong on a well-defined low-dimensional manifold, will the method hallucinate something? **(2)** How well does the method work for more unstructured data?
>
> **(1)** We would like to emphasize that ERDiff isn't constrained to datasets containing a strong 'well-defined' low-dimensional manifold, such as the commonly depicted sphere-like or Swiss-roll-like manifolds. Instead, with the strong expressivity of DM, ERDiff can broadly fit distribution of datasets containing temporal dynamical structures. Meanwhile, it's supported by numerous neural studies [2, 3] that data from various brain regions inherently exhibit temporal dynamical structures. This manifests the wide applicability of ERDiff.
>
> **(2)** If the spatio-temporal structures in data are weak and the time-series resembles a random walk, the alignment task might not benefit as much from ERDiff, which is designed for more structured data. On the other hand, when we aim to apply ERDiff to datasets without a clear trial-structure [4], it is feasible to truncate such continuous data into discrete but meaningful syllables. These syllables typically exhibit clear structural patterns. Then, their overall latent distribution can be learnt through a VAE for subsequent alignment. We conduct additional experiments to validate this approach through an unconstrained rat’s hippocampus dataset [1]. Please refer to Section 1.1 of the general response for result details.
>
> > As the dimensionality of the latent structure increases, how much more badly does the method perform? The authors should quantify this.
>
> In real-world datasets, the dimensionality of the latent structure is inherently fixed. Hence, we increase the dimensionality of latent structure ($\mathbf{z}$) in synthetic datasets and conduct experiments in the following. From the results, we observe that the aligned distribution tends to deviate further from the source-domain distribution as the dimensionality increases. Despite this trend, ERDiff consistently demonstrates superior performance compared to the best baseline method.
>
> |    Method     |      NLL $\downarrow$      |      KLD $\downarrow$      |
> | :-----------: | :------------------------: | :------------------------: |
> | Best-Baseline |     $3.58 (\pm 0.20)$      |     $7.74 (\pm 0.43)$      |
> |   **Ours**    | $\mathbf{3.38} (\pm 0.16)$ | $\mathbf{7.09} (\pm 0.38)$ |
>
> *Table 1: Latent structure dimension $d_z$ = 4*.
>
> |    Method     |      NLL $\downarrow$      |      KLD $\downarrow$       |
> | :-----------: | :------------------------: | :-------------------------: |
> | Best-Baseline |     $6.15 (\pm 0.21)$      |     $12.33 (\pm 0.43)$      |
> |   **Ours**    | $\mathbf{5.53} (\pm 0.29)$ | $\mathbf{11.39} (\pm 0.47)$ |
>
> *Table 2: Latent structure dimension $d_z$ = 8*.
>
> > The authors should address in more detail the kinds of latent structures/time series that would pose more of an issue for their method.
>
> (1) For shorter time-series that may not exhibit strong latent structures, the alignment task becomes less complex. In such cases, even simpler methods like JSDM can have satisfactory results.
>
> (2) If the spatio-temporal structure is weak and the time-series resembles a random walk, the alignment task might not benefit as much from ERDiff, which is designed for more structured data.
>
>
>
> Refs:
>
> [1] Recordings from hippocampal area ca1, pre, during and post novel spatial learning. (Grosmark et al. 2016)
>
> [2] Context-dependent computation by recurrent dynamics in prefrontal cortex. (Valerio et al., 2014)
>
> [3] Network dynamics underlying OFF responses in the auditory cortex. (Giulio et al., 2016)
>
> [4] Hierarchical recurrent state space models reveal discrete and continuous dynamics of neural activity. (Scott et al., 2019)

---

> > ### Comment · Reviewer_1oNu · 2023-08-10
> >
> > Thank you for your detailed response addressing my concerns. Given the response as well as other reviewers comments, I maintain my opinion that the manuscript is suitable for publication and will keep my score at 7.

---

> > > ### Author Response · Authors · 2023-08-11
> > >
> > > We appreciate the timely response.
> > >
> > > Thank you again for your evaluation and recognition of our work.

---

### Official Review · Reviewer_xrSS · 2023-07-04

**Soundness:** 3 good
**Presentation:** 2 fair
**Contribution:** 2 fair
**Rating:** 6
**Confidence:** 3

**Summary:**

Brain-computer interfaces require recalibration to accommodate drifts in the recorded neural populations over time. While there has been some success in aligning neural recordings based on their latent dynamics, deep learning-based alignment methods have also gained popularity since they ignore some of the implicit assumptions made by latent space methods, providing additional flexibility. However, deep learning neural alignment often ignores the temporal structure of the dynamics. The proposed model overcomes these limitations by first extracting the spatio-temporal structure in the source domain via a diffusion model and then aligning the target domain to the source dynamics. The authors demonstrated the success of the method in simulated and neural data, where it outperforms alternative alignment methods.

**Strengths:**

The paper is adequately written and technically sound. The method was tested and shown to work well when applied to a simulated dataset and neural data. The method introduced here uses deep learning-based alignment while still exploiting the temporal dynamics that are critical in neural datasets. Robust alignment of neural recordings is crucial for the success of BCI applications, and in this work, they showed how this method outperforms existing alignment methods in both synthetic and neural datasets. Moreover, they also showed that the alignment can be performed not only across sessions of the same animal but also across animals.

**Weaknesses:**

While the authors show the promise of their method to align neural responses, they overlooked other methods based on the alignment of latent dynamics that have been shown to be successful for BCI applications, as mentioned in the introduction. I believe that a systematic comparison to such methods is critical to fully grasp the significance of this work. For example, CCA, multiset CCA, hyperalignment, or Procrustes-based alignment. In the context of latent space alignment methods, another relevant piece of literature is the method introduced in (https://proceedings.neurips.cc/paper/2021/hash/aad64398a969ec3186800d412fa7ab31-Abstract.html), which also uses neural dynamics for alignment.
Deep learning-based methods allow for more expressive functions, but they often come with additional computational costs, data demands, and the need for careful hyperparameter selection. None of these limitations are addressed in the manuscript, nor is there an explicit comparison across methods (latent space vs. deep learning-based), which could help demonstrate the promise of the method for practical BCI applications. The authors minimally showed the effect of dataset size on performance, but the lowest dataset size tested still had dozens of trials, which could be unrealistic in most practical settings. Additionally, it would be important to report the training times as a function of dataset size, as long training times would render the approach useless for real-time alignment.
The method defines the alignment between a single source and target dataset, but ideally, one would pool data across all sessions for BCI decoding. It would be interesting to note if the proposed method also allows for multi-session alignment.
The authors showed the success of the approach in the context of a single data simulation. However, to fully assess the robustness of the method, they could further test it under different conditions, such as measurement or latent noise, dimensionality, or tasks.


**Questions:**

It is unclear from the text how the training and testing are performed. In the sentence "During testing, we align the test neural data to the training neural data so that we can directly apply the velocity decoder," it is not clear whether the authors include test data for alignment. Additionally, the authors mentioned that behavioral data is used for alignment, but it is also used to evaluate the success of the approach via decoding. This raises the question of whether there is a circular evaluation of the method.
References 9 and 10 cite the preprint and peer-reviewed versions of the same article.


**Limitations:**

The authors should include a section or provide a clear description of the limitations and assumptions of the method. They should also address the computational cost, data demands, and potential implications of the presented work.

---

> ### Author Rebuttal · Authors · 2023-08-10
>
> Thank you for your detailed and constructive comments. We would like to make the following clarifications. Hopefully these will resolve most of your concerns, and they can be taken into account when deciding the final review score.
>
> > They overlooked other methods based on the alignment of latent dynamics that have been shown to be successful for BCI applications. E.g., CCA, multiset CCA, hyperalignment, or Procrustes, ..., another literature is  (amLDS [1]).
>
> We appreciate your suggestion for this comparison. We would like to emphasize that our proposed method, ERDiff, is an **unsupervised** neural distribution alignment method, meaning **no** supervision signals or labels related to behavior are required during the alignment phase. In contrast, the listed methods are **supervised** learning methods. In linear algebra-based methods: CCA and multiset CCA, hyperalignment, and Procrustes-based alignment, they require the knowledge of supervised behavior signals for every trial in both domains to pair the source-target domain trials during the alignment phase. Therefore, these methods are heavily reliant on supervision. Thus, a direct comparison between ERDiff and CCA-based methods may not be a fair evaluation. Regarding amLDS, its probabilistic framework focuses on $K$ discrete stimulus conditions, making it impractical to smoothly expand to our continuous behavioral setting. Therefore, it's also not a direct counterpart to ERDiff.
>
> In practical brain-computer interface (BCI) and neural behavioral applications, unsupervised neural distribution alignment methods are highly desirable. Our proposed ERDiff has the potential to generalize to many more scenarios and fields where supervised signals are not accessible.
>
> > Deep learning-based methods allow for more expressive functions, but they often come with additional computational costs, data demands, and the need for careful hyperparameter selection.
>
> (1) In source domain training phase, the computation cost of ERDiff primarily comes from the diffusion model (DM) training. We note that the DM is trained in the latent space $\mathbf{Z}$, which is significantly smaller than the raw neural signal space $\mathbf{X}$. Therefore, the computational overhead of this one-time training phase is acceptable, especially given that it can be conducted offline in real-world BCI applications.
>
> As for the alignment phase, the computational cost of ERDiff is comparable with those of the baseline methods. More detailed analyses and comparisons of computational costs are provided in Section 1.3 of our global response.
>
> (2) We emphasize that training the diffusion model (DM) used in ERDiff does not require any additional data compared to the baseline methods. The DM uses the same dataset as the baselines for source domain learning as well as distribution alignment.
>
> (3) The hyperparameter selection of our model, including the number of epochs, learning rates, and other relevant parameters, are listed in Section D.1 of our submitted appendix.
>
> > Nor is there an explicit comparison across methods (latent space vs. deep learning-based), which could help demonstrate the promise of the method for practical BCI applications.
>
> We infer the terms 'latent space' and 'deep learning-based' you mentioned might be referring to 'methods specifically designed for neural distribution alignment tasks', and to 'general time-series domain adaptation methods', respectively. Besides their performance comparisons in Table 1 of the manuscript, we additionally conduct comparisons on their computational cost and time complexity, please refer to Section 1.3 of the global response for details.
>
> > The authors minimally showed the effect of dataset size on performance, but the lowest dataset size tested still had dozens of trials
>
> We infer the term 'dataset size tested' you mentioned might be referring to the number of trials used for alignment in the target domain. We would like to highlight that in Fig. 5(B) of the manuscript, we plot the performance of ERDiff and baselines given different numbers of alignment trials. From the plot, it can also be observed that ERDiff maintains a relatively high accuracy when only 25~ trials (representing a 10% sampling density) are used during alignment.
>
> > it would be important to report the training times as a function of dataset size.
>
> Please refer to section 1.3 of the global response for training time cost functions.
>
> > It would be interesting to note if the proposed method also allows for multi-session alignment
>
> Multi-session alignment belongs to the field of multi-domain adaptation (MDA), which is a more complex problem. However, we believe integrating additional components [2] for MDA, ERDiff can generalize to multi-session alignment. We include this direction as part of our future work, listed in section 1.4 of the global response.
>
> > To fully assess the robustness of the method, they could further test it under different conditions, such as measurement or latent noise, dimensionality, or tasks.
>
> Please refer to Fig. 2 in the attached PDF for our robustness study investigating the impacts of latent dimensionality and Gaussian noise on ERDiff.
>
> > it is not clear whether the authors include behavioral data for alignment. This raises the question of whether there is a circular evaluation of the method.
>
> We would like to clarify that no behavioral data or velocity information from the target domain is used in ERDiff. The alignment procedure of ERDiff is performed in an unsupervised manner, showing the broad applicability of our method.
>
> > The authors should include a section or provide a clear description of the limitations and assumptions of the method
>
> Please refer to section 1.4 of the global response for limitations and assumptions.
>
>
>
> Refs:
>
> [1] Across-animal odor decoding by probabilistic manifold alignment. (Pedro et al. 2021)
>
> [2] Multi-Source Unsupervised Domain Adaptation via Pseudo Target Domain.  (Ren et al. 2022)

---

> > ### Comment · Reviewer_xrSS · 2023-08-15
> >
> > I thank the authors for their really comprehensive response. I mostly agree with their comments and I have updated my score accordingly. I still believe that some of there comparisons, even if not tested, should be mentioned in the final version of the manuscript; which also emphasizes the relevance of this method, as they discussed here.

---

> > > ### Author Response · Authors · 2023-08-15
> > > **Thank you**
> > >
> > > We appreciate the reviewer for the kind response and constructive comments. As suggested, we will incorporate the systematic comparison across methods discussed here into the final manuscript.

---

### Official Review · Reviewer_2uTS · 2023-07-07

**Soundness:** 3 good
**Presentation:** 3 good
**Contribution:** 3 good
**Rating:** 7
**Confidence:** 4

**Summary:**

* One of the key challenges in analyzing neural recordings is the scalability of models that link behavior and neural population activity across recording sessions or in inter-subject settings.
• When analyzing single-trial neural population activity, past studies have pointed out that neural activity can be understood in terms of low-dimensional latent dynamics. Such low-dimensional latent dynamics are helpful when visualizing neural profiles across different task conditions and trials.
• Generally, existing methods try to align latent dynamics by minimizing the difference evaluated by the metric between source and target domains. The paper proposes a method to align the source and target domains of multivariate neural data by learning/capturing latent spatiotemporal structure in the source domain with a diffusion model and applying it as a prior on learning/capturing spatiotemporal structure in the target domain.
• The authors applied their model to the non-human primate motor cortex, testing both cross-day and inter-subject recordings.

**Strengths:**

* The authors motivate their approach clearly by arguing that naively aligning time series using domain adaptation is ineffective as
multivariate neural data has low SNR. Thus, leveraging low-dimension representation is a viable option.
• The paper seeks to achieve a form of domain adaptation by aligning the spatiotemporal structure of latent dynamics of the target to the source using a novel alignment method (ERDiff).
• The model and derivations are presented clearly.
• An exhaustive comparison is provided showing that the their model outperforms standard models on both motor cortex and synthetic datasets.

**Weaknesses:**

* The authors must clarify why diffusion models are necessary. How about considerably simpler two-step approaches -- like extracting latents with GPFA (Yu et al, 2009) and aligning them with the proposed ML alignment?
* Alternatively how about comparisons with alternative approaches of comparable complexity -- e.g. adversarial alignment with DANN (Ganin et al, 2015)?

**Questions:**

* Overall: there are many approaches for extracting latent structure from time series data (GPFA, CEBRA, T-PHATE, CILDS) -- one could readily realign the latents extracted from these approaches with the second stage alignment algorithm.
* Apriori, why would one expect the diffusion model to be more effective at aligning the latents than these other approaches?
* Because data from non-human primates is used, please clarify whether appropriate IRB approvals were obtained. Or if this is only a secondary analysis of existing datasets, the approvals obtained in the original studies could be mentioned.

**Limitations:**

* No significant negative societal impact envisioned -- the potential use in BCI may suggest a positive social impact.

---

> ### Author Rebuttal · Authors · 2023-08-10
>
> Thank you for your positive review and insightful questions. We provide clarifications and new results that we have generated to address your questions below. Hopefully these will resolve most of your concerns, and they can be taken into account when deciding the final review score.
>
> > The authors must clarify why diffusion models are necessary. How about considerably simpler two-step approaches -- like extracting latents with GPFA and aligning them with the proposed ML alignment?
>
> This is an insightful question. We would like to emphasize that the diffusion model (DM) is an **essential** and **necessary** component in ERDiff. The traditional methods for identifying low-dimensional latents (e.g., GPFA, LFADS) are **not applicable** to the proposed maximum likelihood alignment (MLA) phase alone. This is because their models only learn a point-to-point mapping function $p(\mathbf{Z} \mid \mathbf{X}^{(s)})$ from neural spike signal to latent factor, rather than explicitly capturing the overall latent distribution $p_s(\mathbf{Z})$ in the source domain. Consequently, in the second step (alignment phase), given the MLA optimization objective: $\underset{\phi}{\operatorname{argmax}} \mathbb{E}_{\mathbf{Z} \sim q\left(\mathbf{Z} \mid \mathbf{X}^{(t)} ; \phi\right)}\left[\log p_s(\mathbf{Z})\right]$ (RHS of Eq.7 in the manuscript), the likelihood $p_s(\mathbf{Z})$ is intractable through those methods alone. In contrast, in our proposed ERDiff, along with the learning of $p(\mathbf{Z} \mid \mathbf{X}^{(s)})$ through VAE, we use DM to learn the overall latent distribution $p_s(\mathbf{Z})$. Therefore, through DM, we can reformulate the MLA objective into Eq. (10) of the manuscript, which is tractable, and computationally efficient.
>
> > Alternatively how about comparisons with alternative approaches of comparable complexity -- e.g. adversarial alignment with DANN (Ganin et al, 2015)?
>
> In the experiment section, the methods SASA, RDF-MMD, and DAF we compared are all of comparable alignment complexity with ours. For a detailed comparison of time complexity, please refer to Section 1.3 of the general response. Here we conducted further experiments (Table 1) using the Domain-Adversarial Neural Network (DANN) method employing 5 random seeds. We implemented the label predictor of DANN with the behavioral velocity predictor. The results of these experiments will be included in the revised manuscript.
>
> |  Method  |       Cross-Day-$R^2$       |       Cross-Day-RMSE       |     Inter-Subject-$R^2$     |     Inter-Subject-RMSE     |
> | :------: | :-------------------------: | :------------------------: | :-------------------------: | :------------------------: |
> |   DANN   |     $-12.57 (\pm 3.28)$     |     $8.28 (\pm 0.32)$      |     $-18.37 (\pm 3.24)$     |     $9.29 (\pm 0.33)$      |
> | **Ours** | $\mathbf{18.81} (\pm 2.24)$ | $\mathbf{7.99} (\pm 0.43)$ | $\mathbf{10.29} (\pm 2.86)$ | $\mathbf{8.34} (\pm 0.34)$ |
>
> *Table 1: Performance Comparison with DANN*
>
> > There are many approaches for extracting latent structure from time series data (GPFA, CEBRA, T-PHATE, CILDS) -- one could readily realign the latents extracted from these approaches with the second stage alignment algorithm.
>
> As we've explained in the first clarification of this response, approaches such as LFADS or GPFA that you mentioned are not applicable for the second MLA stage alone. A diffusion model is necessary to learn an overall $p_s(\mathbf{Z})$ for MLA.
>
> > Apriori, why would one expect the diffusion model to be more effective at aligning the latents than these other approaches?
>
> We appreciate your attention to this important aspect. We would like to clarify that the methods you listed do not serve as neural distribution alignment methods but rather latent variable models (LVM) that are designed to identify latents from raw neural signals.
>
> Besides the learning of an overall $p_s(\mathbf{Z})$ for MLA, we note that the effectiveness of diffusion model (DM) comes from the following two key points.
>
> (1) Precise source domain learning: We note that in behavior-related neural applications, the trial latent dynamics are non-linear and complex. This complexity pose challenges for aligning neural distributions across sessions. Owing to the strong **expressivity** of diffusion models (DM), we first use a DM to learn the overall latent distribution $p_s(\mathbf{Z})$ in source domain. To learn $p_s(\mathbf{Z})$ well, the DM focuses on the spatio-temporal structures of neural latent dynamics in source domain and extracts these structures using the specially designed STBlock. Fig. 2A of the manuscript visualizes the distribution learning and spatio-temporal structure extraction process of the DM. However, previous neural alignment methods ignore these crucial spatio-temporal structures.
>
> (2) Appropriate alignment objective: During the alignment procedure, we propose to use maximum likelihood alignment (MLA) as the optimization objective. We note that this objective aligns well with the DM since the $p_s(\mathbf{Z})$ inside MLA is tractable through DM, and the total MLA formula can be expanded into noise residuals terms that are simple and efficient to compute. Thanks to the stability and flexibility of MLA, the source-domain spatio-temporal structures can be well-recovered in target domain, as illustrated in Fig. 2B and Fig. 4 of the manuscript.
>
> > Please clarify whether appropriate IRB approvals were obtained. Or if this is only a secondary analysis of existing datasets, the approvals obtained in the original studies could be mentioned.
>
> This is an analysis of existing datasets [1] and IRB approvals have been obtained in their studies.
>
>
>
> We look forward to further discussion, and are happy to answer any questions that might arise.
>
> Refs:
>
> [1] Long-term stability of cortical population dynamics underlying consistent behavior. (Lee et al., 2020)

---

> > ### Comment · Reviewer_2uTS · 2023-08-20
> >
> > I thank the authors for the detailed clarifications and additional experiments, and have updated my score accordingly. I have no further questions for the authors.

---

### Official Review · Reviewer_wx95 · 2023-07-13

**Soundness:** 4 excellent
**Presentation:** 3 good
**Contribution:** 3 good
**Rating:** 7
**Confidence:** 2

**Summary:**

This paper proposes a distribution alignment method (ERDiff), which combines extraction of spatio-temporal structures in latent dynamics from the source distribution and maximum likelihood alignment procedure on the target domain.  The proposed method was evaluated on both synthetic and real data (neural recordings from non-human primate motor cortex), outperforms other methods under both cross-day and inter-subject settings.


**Strengths:**

- The proposed method is novel and is technically sold
- Performance of the method is well demonstrated on real data under inter-session/subject setup suggest that it could be an important tool with potential broad use in many field not just in neuroscience.


**Weaknesses:**

-I don’t have any major concerns. Although the methods has been shown to outperform some of the current techniques, advantage of the proposed approach is not well demonstrated. I would like to see some analysis on computational cost


**Questions:**

- Although the authors clearly mentioned the limitation of alignment method based on pre-defined metric, it would be nice see how the proposed method performs compared to these.


**Limitations:**

- Limitations is not addressed in the draft

---

> ### Author Rebuttal · Authors · 2023-08-10
>
> Thank you for your positive review and insightful questions. We provide clarifications and new results that we have generated to address your questions below.
>
> > Although the methods have been shown to outperform some of the current techniques, advantage of the proposed approach is not well demonstrated.
>
> We appreciate your attention to this important aspect. We'd like to emphasize that the advantages of our proposed ERDiff in neural distribution alignment comes from the following two key points.
>
> **(1)** Precise source domain learning: We note that in behavior-related neural applications (e.g., brain-computer interfaces), the trial latent dynamics are non-linear and complex. This complexity pose challenges when it comes to aligning neural distributions across sessions (domains). Owing to the strong **expressivity** of diffusion models (DM), we first use a DM to learn the overall latent distribution $p_s(\mathbf{Z})$ in source domain. To learn $p_s(\mathbf{Z})$ well, the DM focuses on the spatio-temporal structures of neural latent dynamics in source domain and extracts these structures using the specially designed STBlock. Fig. 2A of the manuscript visualizes the distribution learning and spatio-temporal structure extraction process of the DM in source domain. However, previous neural alignment methods [1, 2] ignore these crucial spatio-temporal structures.
>
> **(2)** Appropriate alignment objective: During the alignment procedure, we propose to use maximum likelihood alignment (MLA) as the optimization objective. We note that this objective aligns well with the DM since the $p_s(\mathbf{Z})$ inside MLA is tractable through DM, and the entire MLA formula can be expanded into noise residuals terms that are simple and efficient to compute. Thanks to the stability and flexibility of MLA, the source-domain spatio-temporal structures can be well-recovered in target domain, as illustrated in Fig. 2B and Fig. 4 of the manuscript. In contrast, methods based on pre-defined metrics often make a strong assumption that the overall latent distributions follow a Gaussian, which significantly restricts the expressiveness of those methods. For methods based on adversarial training, even though their optimization objectives can be formed into Jensen–Shannon divergence (JSD), their practical training steps always lack stability [3], and are far from reaching the optimization target (JSD).
>
> > I would like to see some analysis on computational cost.
>
> In source domain training phase, the additional computation cost of ERDiff primarily comes from the diffusion model (DM) training. We note that the DM is trained in the latent space $\mathbf{Z}$, which is significantly smaller than the raw neural signal space $\mathbf{X}$ in magnitude. Therefore, the computational overhead of this one-time training phase is acceptable, especially given that it can be conducted offline in real-world BCI applications.
>
> In the alignment phase, we would like to emphasize that ERDiff maintains a comparable computational cost with baseline methods. Please refer to Section 1.3 of the global response for a comprehensive analysis of the computational cost and time complexity.
>
> > Although the authors clearly mentioned the limitation of alignment method based on pre-defined metric, it would be nice see how the proposed method performs compared to these.
>
> In the experiment section of the manuscript, among the group of methods based on pre-defined metrics, we show the results of JSDM since it achieves the highest performance. To provide a more thorough comparison, here we additionally conduct experiments with two more methods in this group: Kullback–Leibler divergence minimization (KLDM) and Wasserstein distance minimization (WDM). These two metrics form the backbone of previous methods on neural distribution alignment [4] and [5], respectively. We can observe that ERDiff consistently demonstrates superior performance compared to these two methods. These additional comparative results will be incorporated into the revised manuscript.
>
> |  Method  |       Cross-Day-$R^2$       |       Cross-Day-RMSE       |     Inter-Subject-$R^2$     |     Inter-Subject-RMSE     |
> | :------: | :-------------------------: | :------------------------: | :-------------------------: | :------------------------: |
> |   KLDM   |     $-31.63 (\pm 1.85)$     |     $10.55 (\pm 0.37)$     |     $-30.79 (\pm 2.24)$     |     $10.42 (\pm 0.37)$     |
> |   WDM    |     $-19.85 (\pm 2.45)$     |     $8.61 (\pm 0.36)$      |     $-17.74 (\pm 2.71)$     |     $9.21 (\pm 0.43)$      |
> | **Ours** | $\mathbf{18.81} (\pm 2.24)$ | $\mathbf{7.99} (\pm 0.43)$ | $\mathbf{10.29} (\pm 2.86)$ | $\mathbf{8.34} (\pm 0.34)$ |
>
> *Table 1: Performance comparison with alignment methods based on pre-defined metrics*
>
> > Limitations is not addressed in the draft
>
> We delve into the Limitations, Future Work, and Broader Impact of our work. (Please refer to Section 1.4 of the global response for details.) This comprehensive discussion will be put into the revised manuscript in a new 'Discussion' section, including and replacing the existing 'Section 5: Conclusion'.
>
>
>
> We look forward to further discussion, and are happy to answer any questions that might arise.
>
> Refs:
>
> [1] Robust alignment of cross-session recordings of neural population activity. (Justin et al., 2022)
>
> [2] Stabilizing brain-computer interfaces through alignment of latent dynamics. (Brianna et al. 2022)
>
> [3] Evaluation of Mode Collapse in Generative Adversarial Networks. (Sayeri et al., 2018)
>
> [4] Stabilizing brain-computer interfaces through alignment of latent dynamics. (Brianna et al. 2022)
>
> [5] Hierarchical Optimal Transport for Multimodal Distribution Alignment. (John et al. 2019)

---

> > ### Comment · Reviewer_wx95 · 2023-08-17
> >
> > I appreciate the authors for addressing my questions in detail. I will keep my original score at 7.

---

### Official Review · Reviewer_tfQ5 · 2023-07-13

**Soundness:** 3 good
**Presentation:** 4 excellent
**Contribution:** 2 fair
**Rating:** 6
**Confidence:** 2

**Summary:**

The paper focuses on aligning highly variable neural population activities across days and subjects to stabilize the learning proposes and advance applications such as the brain computer interface. The idea is to train a VAE on one dataset that is hopefully self-consistent and then using a VAE trained on a dataset from another day or subject align the conditional distributions (encoders) for the latent spaces maximizing the likelihood of the source domain latent space under the new encoder distribution. The difficulty is the need to model the marginal source distribution of the latent space, which the work does using a diffusion model. The approach is demonstrated in comparison with alternative models on a synthetic dataset and actual neural recordings.


**Strengths:**

1. A well written paper (but the abstract).
2. An interesting application of diffusion models.
3. Potentially impactful in practice of the BCI, more work, including further evaluation, is needed here however.


**Weaknesses:**

1. A niche application and demonstration. A cellular neuroscience focused paper with no additional effort to demonstrate a general applicability of the approach in evaluations.
2. Poorly written abstract, especially in contrast to the rest of the paper.


**Questions:**

1. Is the code going to be released publicly? Looks like the success of the approach depends less on the high level probabilistic descriptions in the paper than on the details of the actual implementation.
2. Is the data going to be released publicly for reproducibility?


**Limitations:**

Potentially, the applicability of this work may be limited only to the demonstrated use in intracranial multicellular recordings, and it may not contribute to advancements in other areas of Machine Learning (ML). No demonstration was provided to counter this potential limitation.

---

> ### Author Rebuttal · Authors · 2023-08-10
>
> Thank you for your valuable comments. We would like to make the following clarifications. Hopefully these will resolve most of your concerns, and they can be taken into account when deciding the final review score.
>
> > A cellular neuroscience focused paper with no additional effort to demonstrate a general applicability of the approach in evaluations. Potentially, the applicability of this work may be limited only to the demonstrated use in intracranial multicellular recordings, and it may not contribute to advancements in other areas of Machine Learning (ML).
>
> We would like to emphasize that the focus on this work lies in behavioral neuroscience research and its related applications (e.g., brain-computer interfaces (BCI)) and we have selected "Neuroscience and cognitive science" as the primary area for this paper. Within the field of neuroscience, we note that the focus of this paper (i.e., neural distribution alignment) is of vital importance. Therefore, we mainly validate the effectiveness of the proposed ERDiff on various neural recording datasets.
>
> On the other hand, we fully agree that the generalizability of each method is an important consideration. Therefore, we have conducted experiments on synthetic datasets (Section 4.1 of the manuscript) in which more general time-series data are simulated. Moreover, in Section 1.1 of the general response, we also provide additional experimental results on an unconstrained rat’s hippocampus dataset [1]. These results prove the performance enhancement by ERDiff in a more general context and will be provided in our revised manuscript.
>
> Here we further discuss the contribution of ERDiff to the broader Machine Learning field:
>
> **(1)** In the field of domain adaptation, we propose to use diffusion model (DM) and STBlocks (Fig.2(A) of the manuscript) to extract spatio-temporal structures of the source domain distribution for later alignment propose. We note that the preservation of spatio-temporal structures is a key component for accurate adaptation, and such structures are ubiquitous in dynamical time-series data outside the neuroscience field [2, 3].
>
> **(2)** We propose the maximum likelihood alignment (MLA). The robust statistical properties of MLA ensure stability during the alignment phase and offer broad applicability across a range of time-series datasets. We also note that the optimization objective of MLA aligns well with the DM since the $p_s(\mathbf{Z})$ inside MLA is tractable through DM, and the entire MLA formula can be expanded into noise residuals terms that are simple and efficient to compute.
>
> Hence, we believe ERDiff has potential use in underlying structure extraction and domain adaptation tasks of general time-series data (e.g., weather forecasting [2] and seismology [3]). We appreciate that Reviewer 9MfT, wx95, and 1oNu have recognized the potential generalizability of ERDiff. In the revised manuscript, we plan to incorporate the aforementioned insights into the broader impact part of ERDiff (please refer to Section 1.4 of the general response).
>
> > Poorly written abstract, especially in contrast to the rest of the paper.
>
> We thank the reviewer for this practical suggestion. The following is our re-written abstract and we will update it in our revised version of the manuscript.
>
> "In the field of behavior-related brain computation, it is necessary to align raw neural signals against the drastic domain shift among them. A foundational framework within neuroscience research assumes that trial-based neural activities rely on low-dimensional latent dynamics. Focusing on such latent dynamics greatly assists the alignment procedure. Despite the great progress the field have reached, existing methods usually ignore the intrinsic spatio-temporal structures during alignment. Thus, those solutions lead to poor quality in dynamics structures and overall performance after alignment. To tackle this problem, we propose a method leveraging the expressivity of diffusion model to relieve such issues. Specifically, the latent dynamics structures of the source domain are first extracted by the diffusion model. Then, such structures are well-recovered through a maximum likelihood alignment procedure in the target domain. We first demonstrate the effectiveness of our proposed method on a synthetic dataset. Then, when applied to neural recordings from primate motor cortex, under both cross-day and inter-subject settings, our method consistently manifests its capability of preserving the spatio-temporal structure of latent dynamics and outperforms existing approaches in alignment quality."
>
> > Is the code going to be released publicly? Looks like the success of the approach depends less on the high level probabilistic descriptions in the paper than on the details of the actual implementation.
>
> Yes. We would release the codes in the supplementary material to public. The detailed implementations of ERDiff on all datasets are described in section A.1 and D.1 of the appendix. To further ensure the reproducibility of our experimental results, we will provide a more comprehensive listing of these implementation details in the appendix of our revised manuscript.
>
> > Is the data going to be released publicly for reproducibility?
>
> Yes. The dataset is collected by [4] and it is available from the corresponding author upon reasonable request.
>
>
>
> We look forward to further discussion, and are happy to answer any questions that might arise.
>
>
>
> Refs:
>
> [1] Recordings from hippocampal area ca1, pre, during and post novel spatial learning. (Grosmark et al.,2016).
>
> [2] Application of Domain Adaptation Approach for Weather Data Mining. (Yang et al., 2018)
>
> [3] Seismic Facies Analysis: A Deep Domain Adaptation Approach. (Quamer et al., 2021)
>
> [4] Long-term stability of cortical population dynamics underlying consistent behavior. (Lee et al., 2020)

---

> > ### Comment · Reviewer_tfQ5 · 2023-08-13
> >
> > Thank you for your explanations and an improved abstract. I still hold that a generality can only be demonstrated in a wider set of experiments rather than hypothesized. I do value the potential uses that a method like the proposed can have if it works outside of the demonstrated domain, however as it stands, the evidence that it does is lacking.
> >
> > Nevertheless, this is a strong manuscript and an interesting approach that fits the "Neuroscience and cognitive science" section.

---

> > > ### Author Response · Authors · 2023-08-13
> > > **Additional Experiments on General Time-series Domain-Adaptation Datasets**
> > >
> > > We thank the reviewer for the valuable response. We truly agree that the evidence from experiments carries more weights in determining the generalizability of each method. Here we additional conduct experiments on two general time-series datasets widely used in domain-adaptation papers:
> > >
> > > (1) Boiler Fault Detection Dataset [1]. The dataset contains sensor data from three distinct boilers, collected between March 24, 2014, and November 30, 2016. Each boiler in this dataset is considered as a unique domain  (we represent as 1,2, and 3). The objective of the learning task is to predict the faulty blowdown valve of each boiler. The results can be found in Table 1 below.
> > >
> > > (2) City Air Quality Forecast Dataset [2]. The dataset is composed of air quality, meteorological, and weather forecast data from three cities, denoted as A, B, and C. Each city is treated as a unique domain. Using both the air quality and meteorological data, our objective is to predict PM2.5 levels. The results can be found in Table 2 below.
> > >
> > > *Implementation Details:* Besides the three methods that focus on general time-series domain adaptation we compared in the manuscript, we add one more fundamental baseline: LSTM_S2T. This approach trains a vanilla LSTM model using source domain data and then directly applies it to the target domain without adaptation. This method represents the performance lower bound. In ERDiff, we apply $4$ STBlocks in the diffusion model (DM). For a fair comparison, we set the size of the latent dimension equal to the representation space size used in other methods. The batch size is set as 128.
> > >
> > > Owing to the strong domain learning capabilities of DM and our proposed corresponding maximum likelihood alignment (MLA) in adaptation phase, most times ERDiff outperforms existing methods in terms of alignment quality and it reaches the highest performance on average.
> > >
> > > |   Method   | 1$\rightarrow$2 | 1$\rightarrow$3 | 3$\rightarrow$1 | 3$\rightarrow$2 | 2$\rightarrow$1 | 2$\rightarrow$3 |    Avg    |
> > > | :--------: | :-------------: | :-------------: | :-------------: | :-------------: | :-------------: | :-------------: | :-------: |
> > > |  LSTM_S2T  |      67.09      |      94.54      |      93.14      |      56.09      |      84.99      |      91.31      |   81.19   |
> > > |    SASA    |      71.54      |      96.39      |    **94.77**    |      63.15      |      87.76      |      93.59      |   84.53   |
> > > |  RDA-MMD   |      73.95      |      96.30      |      94.14      |      65.05      |      88.11      |    **94.42**    |   85.34   |
> > > |    DAF     |      74.55      |   **96.54***    |      94.58      |      65.03      |      88.85      |      94.19      |   85.59   |
> > > | **ERDiff** |   **75.26***    |      96.13      |      94.14      |   **66.66***    |   **89.09***    |      94.02      | **86.21** |
> > >
> > > *Table 1: AUC Score ($\%$) on Boiler Fault Detection Dataset. $\star$ denotes significance p-value <0.02 compared with the best baseline.*
> > >
> > >
> > >
> > > |   Method   | B$\rightarrow$A | C$\rightarrow$A | A$\rightarrow$B | C$\rightarrow$B | B$\rightarrow$C | A$\rightarrow$C |    Avg    |
> > > | :--------: | :-------------: | :-------------: | :-------------: | :-------------: | :-------------: | :-------------: | :-------: |
> > > |  LSTM_S2T  |      40.20      |      48.91      |      52.81      |      68.14      |      13.82      |      13.82      |   39.62   |
> > > |    SASA    |      34.26      |      40.91      |      48.15      |      56.80      |      13.49      |      13.46      |   34.51   |
> > > |  RDA-MMD   |      32.98      |      37.88      |      45.42      |      52.78      |    **13.19**    |      13.18      |   32.57   |
> > > |    DAF     |      31.75      |      36.86      |      44.24      |      52.93      |      13.22      |      13.07      |   32.02   |
> > > | **ERDiff** |   **31.05***    |   **35.45***    |    **43.30**    |   **51.36***    |      13.41      |    **13.03**    | **31.28** |
> > >
> > > *Table 2: RMSE on Cities Air Quality Forecast Dataset.  $\star$ denotes significance p-value <0.02 compared with the best baseline.*
> > >
> > >
> > >
> > > We thank the reviewer for the kind comments.
> > >
> > >
> > >
> > > [1] used in: Time Series Domain Adaptation via Sparse Associative Structure Alignment. (Ruichu et al., 2021)
> > >
> > > [2] https://www.microsoft.com/en-us/research/project/urban-air/

---

### Official Review · Reviewer_9MfT · 2023-07-18

**Soundness:** 3 good
**Presentation:** 2 fair
**Contribution:** 3 good
**Rating:** 6
**Confidence:** 3

**Summary:**

Inter-individual and inter-session variability significantly complicate direct comparison of neural recordings collected over time, degrading trained behavioral models. This can be cast as a more general distribution alignment problem, shared across unsupervised learning. To address neural distribution alignment, the authors introduce a novel "ERDiff" method, which co-trains a variational autoencoder and a diffusion model to extract latent spatiotemporal structure in a source dataset. To align with a desired target dataset, parameter finetuning is performed on the read-in layer of the probabilistic encoder learned during training to match the target dataset.
Simulation and experimental results suggest that ERDiff captures relevant spatiotemporal structure, performing competitively against other baseline methods including those based on overall minimizing distribution divergence as well as based on adversarial methods. Overall, the authors argue that ERDiff is better able to capture the important spatiotemporal structure of their trialwise data; for example, from a monkey center-out reaching task in the experimental results. This focus differentiates ERDiff from many other alignment methods, which ignore unfolding dynamics in their alignments.

**Strengths:**

Incorporating spatiotemporal structure into the alignment of neural recordings is a novel approach, as these methods traditionally consider successive data points as independent samples or learn a set of low-dimensional latent dynamics which can then be aligned. These dynamics are particularly critical where the the trialwise dynamics strongly influence both behavior and neural activity over time. By directly learning and aligning the latent spatiotemporal structure, ERDiff shows stable performance even over relatively low sampling density, retaining relatively high decoding performance compared to other baseline methods even with decreasing numbers of trials in the target domain. These properties suggest that it is also likely that the general ERDiff approach may be useful in other cases where distribution shift between a source and target domain obscures but does not remove a shared latent structure.
The general approach of combining variational autoencoders (VAEs) and diffusion models (DMs) has been previously introduced (e.g., Panday et al., TMLR, 2023); however, ERDiff is a significantly different formulation of the idea and represents a novel approach in leveraging the relative strengths of these methods through cooperative training.

**Weaknesses:**

While the current experiments extensively compare inter-session and inter-subject differences in real neural recordings --- in addition to the simulated experiments --- it is not clear to what extent the presented findings might generalize to data sources without such a clear trial structure. For example, in recordings collected during unconstrained exploration or sleep, low-dimensional latent factors may still drive a successful alignment. Nonetheless, it is not clear how ERDiff would best be deployed in that context. This is particularly relevant as the described real-data experiments additionally incorporated velocity information, and the performance of ERDiff without a behavioral signal during training is thus unclear.
In the current paper, my additional concern with the experiments is on the relative baselines against which ERDiff is compared. It would be informative to see a direct comparison with canonical correlation analysis (CCA) approaches, which have been used to date in aligning neural datasets with a strong temporal correspondence (e.g., Gallego et al., Nat Neuro, 2020).

**Questions:**

Would the authors be able to directly comment on the relationship between their work and other field-standard methods to identify low-dimensional, latent factors such as LFADS (Pandarinath et al., Nat Methods, 2018) ? In particular, the relative benefits of learning the low-dimensional latent structure as part of the alignment --- as compared to other existing methods which learn low-dimensional dynamics which can then be aligned --- is not clearly explained. This would help to better situate the work in the literature.

**Limitations:**

Although I do not see a Broader Impact section included in the current submission, I do not anticipate potential negative societal impacts given the constrained focus of the work.

---

> ### Author Rebuttal · Authors · 2023-08-10
>
> Thank you for your detailed and constructive comments. We would like to make the following clarifications. Hopefully these will resolve most of your concerns, and they can be taken into account when deciding the final review score.
>
> > **(1)** It is not clear to what extent the presented findings might generalize to data sources without such a clear trial structure. **(2)** This is particularly relevant as the described real-data experiments additionally incorporated velocity information, and the performance of ERDiff without a behavioral signal during training is thus unclear.
>
> **(1)** This is a highly valid point. We would like to emphasize that the target of performing alignment between neural distributions is to adapt the neural-behavior mapping function from the source domain to the target domain. The consistency of such mapping function within every single domain is a prerequisite. Therefore, our paper and previous works [1, 2] in neural distribution alignment mainly focus on datasets where behaviors have been confirmed to maintain a consistent trial-wise dynamical structure.
>
> On the other hand, when we aim to apply neural distribution alignment (i.e., ERDiff) to datasets without a clear trial structure [3,4], it is feasible to truncate such continuous behavioral data into discrete but meaningful segments and syllables (e.g., grooming, running). These behavioral segments typically exhibit clear structural patterns. Then, their overall latent distribution can be learnt through a VAE for subsequent alignment. We conduct additional experiments to validate this approach through an unconstrained rat’s hippocampus dataset [5]. Please refer to Section 1.1 of the general response for details.
>
> **(2)** Please refer to Section 1.2 of the global response for a detailed discussion of this point.
>
> > It would be informative to see a direct comparison with canonical correlation analysis (CCA) approaches
>
> We notice that CCA is a well-known method for neural distribution alignment. However, practical approaches based on CCA [6] require supervised behavior signals of the target domain to pair the source-target domain trials during the alignment phase. In contrast, our proposed ERDiff is an **unsupervised** neural distribution alignment method, having the potential to generalize to broader datasets and applications where supervised behavior signals are inaccessible. Here, for a fair comparison, we remove the behavior signals in practical CCA approach [6] and report the results in the following:
>
> |  Method  |       Cross-Day-$R^2$       |       Cross-Day-RMSE       |     Inter-Subject-$R^2$     |     Inter-Subject-RMSE     |
> | :------: | :-------------------------: | :------------------------: | :-------------------------: | :------------------------: |
> |  U-CCA   |     $-25.56 (\pm 2.13)$     |     $9.70 (\pm 0.28)$      |     $-29.26 (\pm 2.11)$     |     $10.43 (\pm 0.46)$     |
> | **Ours** | $\mathbf{18.81} (\pm 2.24)$ | $\mathbf{7.99} (\pm 0.43)$ | $\mathbf{10.29} (\pm 2.86)$ | $\mathbf{8.34} (\pm 0.34)$ |
>
> *Table 1: Performance comparison with Unsupervised-CCA*
>
> > **(1)** Would the authors be able to directly comment on the relationship between their work and other field-standard methods to identify low-dimensional, latent factors such as LFADS ? **(2)** The relative benefits of learning the low-dimensional latent structure as part of the alignment --- as compared to other existing methods which learn low-dimensional dynamics which can then be aligned --- is not clearly explained.
>
> **(1)** This is a good point. While Latent Variable Models (LVM) such as LFADS are focused on identifying low-dimensional, latent factors from raw neural signals, the primary goal of ERDiff is quite different. ERDiff aims to perform unsupervised neural distribution alignment (domain adaptation) given these identified latent factors in the source and target domains. On the other hand, not limited to the current canonical VAE, ERDiff has the potential to perform alignment based on latent factors identified by alternative LVMs (e.g., LFADS). We thus include this topic as part of our future work (please refer to Section 1.4 of the global response).
>
> **(2)** We apologize for the confusion. In ERDiff, we would like to clarify that the learning of low-dimensional latents is **separate** from the alignment. In the outlined cooperative training procedure, during each iteration, the DM uses the inferred latents by the VAE as input data. However, meanwhile, DM does not affect the VAE's learning. This means the procedure of learning the low-dimensional latents is disentangled and separate from the procedure of DM learning and neural distribution alignment.
>
> We note that the cooperative training of VAE and diffusion model (DM) assists the DM in accurately extracting the distribution of the source domain latents, which ultimately improves the alignment performance. This is why we include a description of the VAE training (learning the low-dimensional latent) phase in the source domain.
>
> > I do not see a Broader Impact section included in the current submission.
>
> Please refer to Section 1.4 of the global response.
>
>
>
> We look forward to further discussion, and are happy to answer any questions that might arise.
>
> Refs:
>
> [1] Robust alignment of cross-session recordings of neural population activity by behaviour via unsupervised domain adaptation. (Justin et al., 2022)
>
> [2] Stabilizing brain-computer interfaces through alignment of latent dynamics. (Brianna et al. 2022)
>
> [3] Hierarchical recurrent state space models reveal discrete and continuous dynamics of neural activity. (Scott et al., 2019)
>
> [4] The Striatum Organizes 3D Behavior via Moment-to-Moment Action Selection. (Jeffrey et al., 2018)
>
> [5] Recordings from hippocampal area ca1, pre, during and post novel spatial learning. (Grosmark et al.,2016)
>
> [6] Long-term stability of cortical population dynamics underlying consistent behavior. (Gallego et al., 2020)

---

> > ### Comment · Reviewer_9MfT · 2023-08-17
> >
> > I thank the authors for their comprehensive response in clarifying the contribution of this work, and I’ve updated my score correspondingly.
> >
> > In particular, I find the attentional experiments on (1) a rat hippocampus dataset and (2) when removing the behavioral signals from ERDiff particularly compelling. These broadly reinforce the author’s point that a trialwise structure—though not behavioral information—is necessary for a successful application of ERDiff.
> >
> > I appreciate the direct comparison with unsupervised-CCA, but I am still uncertain that this is the right baseline. Would it not be more meaningful to compare unsupervised-CCA with the ERDiff model without behavior signals of source domain during VAE training?

---

> > > ### Author Response · Authors · 2023-08-18
> > > **Thank you**
> > >
> > > We sincerely appreciate the reviewer's positive evaluation of our additional experiments and contribution. We apologize for the ambiguity and would like to clarify that the term 'unsupervised' in unsupervised-CCA denotes unsupervised neural distribution alignment. This refers to the exclusion of supervised behavioral labels (e.g., direction and velocity) in the **target domain** during the alignment phase. In the above experiments of unsupervised-CCA (U-CCA) we presented, the behavioral signals from the **source domain** are incorporated during VAE training. Hence, we compare it with the version of ERDiff that also incorporates behavior signals of source domain. We thank the reviewer once more for the valuable response and suggestions.

---

### Author Rebuttal · Authors · 2023-08-09

We would like to express our sincere gratitude to all the reviewers for their insightful feedback and suggestions. We appreciate the positive comments which characterized our work as having a `"clear motivation"` (2uTS), `"novel"` methodological progression (9MfT, wx95), being `"technically solid"` (wx95, xrSS), with a `"well-written and clear presentation"` (tfQ5, 2uTS), conducting `"exhaustive comparison"` in the experiment (2uTS) and recognized our work to have `"potential broad use"` (wx95, 1oNu). In this place, we would like to first provide several general clarifications to enhance overall understanding of our work.

**1.1 Generalizability of ERDiff across Datasets**

The spatio-temporal structures are intrinsic to neural dynamics associated with behaviors, and have been extensively studied in neuroscience [1,2]. We would like to emphasize that with the powerful distribution learning ability of diffusion model (DM), ERDiff has great potential to generalize across a wide range of neural behavior datasets and applications. For validation, we run additional experiments on an unconstrained rat’s hippocampus dataset [3] and verify the efficacy of ERDiff on it (please refer to Fig. 1 and Table 1 of the attached PDF). These results will be provided in the revised manuscript.

**1.2 ERDiff is an unsupervised neural distribution alignment method**

To be in line with the conventions of previous studies on neural distribution alignment [4, 5], in the manuscript, we agree that behavioral (velocity) signals of the source domain are present during the VAE training. These signals do help in learning a more interpretable neural latent space. However, we emphasize that ERDiff does not incorporate any behavioral signals of the target domain during the distribution alignment phase. Hence, ERDiff is entirely an **unsupervised** neural distribution alignment (i.e., test-time adaptation) method.

We also note that the introduction of behavior signals in the source domain is an alternative choice. We conduct additional experiments to verify the efficacy of ERDiff when such behavioral signals are removed. Please refer to Table 3 of the attached PDF for detailed results.

**1.3 (1) Computational Cost and (2) Time Complexity of ERDiff of Alignment Phase**

**(1)** In the alignment phase, for any given target domain, ERDiff can stably align it to the source domain in a comparable overhead with baselines. In Table 2 of the attached PDF, we conduct a comparative analysis between ERDiff and baseline methods in terms of additional parameter number, additional model size, stability, and alignment time. The demonstrated alignment time corresponds to the execution time for aligning one iteration (a batch of size 64) on a MacBook Pro (2019 equipped with 8-Core Intel Core i9 and 4 GB RAM). These analyses will be provided in the revised manuscript.

**(2)** Here we conduct time complexity analysis with respect to the batch size $B$ for the alignment phase. The ERDiff's alignment objective is composed of two main terms: Diffusion Noise Residual and Sinkhorn Divergence. We note that in the diffusion noise residual computation, it does not go through the entire $T$ diffusion steps. Instead, it just samples a single time step (noise scale) $t$ and calculates the noise residual specific to that step. Thus, the total complexity of this part takes $\mathcal{O}(K_1 *B * d)$, in which the coefficient $K_1$ relates to the inference complexity of the DM denoiser $\boldsymbol{\epsilon}\left(\mathbf{Z}, t \right)$; $d$ denotes the latent dimension size. For the Sinkhorn Divergence, it has to compute the distance matrix, costing $\mathcal{O}(K_2 *B^2)$; $K_2$ is a relatively small coefficient in magnitude. By summing up, the total complexity of ERDiff is given by $\mathcal{O}(K_1 *B * d + K_2 *B^2)$. This $\mathcal{O}(B^2)$ complexity is applicable since the non-adversarial baseline methods we compared (i.e., JSDM, and SASA) require quadratic complexities as well.

These analyses will be provided in the revised manuscript.

**1.4  New 'Discussion' Section**

Here we delve into Limitations, Future Work, and Broader Impact of our work. These parts will be put into the revised manuscript in a new 'Discussion' section, including and replacing the existing 'Section 5: Conclusion'.

*Limitation and Future Work*: **(1)** Multi-domain Adaptation. Currently, ERDiff can align well with a single source domain latent distribution. An intriguing direction for future work would be learning a unified latent space across multiple source domains using the diffusion model. Thus the method would be applicable to domain generalization problems. **(2)** Generalization on alternative latent variable models (LVM). ERDiff currently identify the latent variables of raw neural signals with a canonical version of VAE. However, the architecture of the LVM within ERDiff is actually disentangled from the diffusion model training or MLA procedure. Future work includes validating ERDiff given multiple implementations of LVM (e.g., LFADS, pi-VAE).

*Broader Impact:* Not confined to computational neuroscience, the cooperative training technique and the MLA in ERDiff have potential to apply into broader domain adaptation tasks across general time-series datasets (e.g., weather forecasting, and seismology). We also expect that our method can be applied or extended to other BCI applications and the broader field of neuroscience/AI.

Refs:

[1] STNDT: Modeling Neural Population Activity with Spatiotemporal Transformers. (Trung et al. 2022)

[2] Deep inference of latent dynamics with spatio-temporal super-resolution. (Feng et al. 2021)

[3] Recordings from hippocampal area ca1, pre, during and post novel spatial learning. (Grosmark et al. 2016)

[4] Robust alignment of cross-session recordings of neural population activity. (Justin et al., 2022)

[5] Stabilizing brain-computer interfaces through alignment of latent dynamics. (Brianna et al. 2022)

---

### Decision · Program_Chairs · 2023-09-21

**Decision:**

Accept (spotlight)

**Comment:**

The reviewers agree that this is a novel, clear, and technically solid contribution. The additional work during the rebuttal and discussion phase further improved the manuscript. There is a growing need for methods that can align neural and behavioral activity across individuals and sessions, and ERDiff shows good potential for tackling this problem.